# Transfer Entropy as a Measure of Information Flow in LLMs and VLMs

## Abstract

Understanding how information propagates within large language models (LLMs) and vision-language models (VLMs) is central to interpretability and efficient deployment. Attention weights indicate where probability mass is allocated but do not quantify directional influence or redundancy across layers and heads. We introduce *transfer entropy* (TE) as an information-theoretic measure of directed information flow in Transformers, and develop tractable TE estimators for high-dimensional hidden states. We analyze unimodal (RoBERTa, T5, Llama-3.2-3B, Qwen-2.5-7B) and multimodal (CLIP ViT-B/16, LLaVA-1.5-7B) models, finding consistent depth-dependent TE structure across architectures. Encoder-only RoBERTa exhibits a mid-layer peak, while decoder-only LLMs concentrate computation in a broad mid-depth band with early/late layers acting as input/output adapters. In T5, encoder and decoder TE diverge: the encoder remains uniformly high whereas the decoder peaks mid-depth. For CLIP, redundancy clusters in late vision blocks while the text tower maintains elevated TE. In LLaVA-1.5-7B, vision→text TE increases with depth, suggesting multimodal fusion is localized to upper decoder layers. These layerwise TE profiles expose depth-dependent redundancy, enabling targeted pruning with minimal accuracy loss. At the head level, combining TE with CLS/EOT attention mass supports role-aware head classification, yielding interpretable responsibility maps across depth. Overall, TE provides a unified lens for diagnosing, interpreting, and compressing Transformer-based models.

## 1. Introduction

Large Language Models (LLMs) and Vision–Language Models (VLMs) have become central to modern AI, driving breakthroughs in natural language understanding, visual recognition, and multimodal reasoning. Their success is largely attributable to the Transformer architecture, whose self-attention layers enable flexible integration of contextual information across sequences and modalities. Despite these advances, the internal information dynamics of Transformers remain opaque: it is often unclear how different layers, tokens, and attention heads actually contribute to the model's representational updates. Attention weights reveal where probability mass is allocated, but they cannot distinguish causal influence from incidental correlation, leaving a gap in interpretability and compression.

To address this, we study *transfer entropy* (TE) as a principled measure of directional information flow within Transformers. TE, grounded in information theory, quantifies how much the future state of one process becomes predictable when conditioned on the present state of another, beyond what is explained by its own history. Applied to LLMs and VLMs, TE captures how much of a layer's, token's, or head's new representation is attributable to its neighbors, thereby separating true causal influence from spurious attention. Unlike static probes or gradient-based attributions, TE tracks dynamics across depth and training, exposing redundancy, bottlenecks, and zones of specialization.

In this work, we develop tractable estimators of TE tailored to high-dimensional model states and use them to profile information flow in both unimodal (RoBERTa, T5, Llama-3.2-3B, Qwen-2.5-7B) and multimodal (CLIP ViT-B/16, LLaVA-1.5-7B) architectures, revealing consistent depth-dependent structure across models: encoder-only networks peak sharply in the middle layers; decoder-only LLMs concentrate computation in a broad mid-layer band; encoder-decoder models such as T5 show divergent TE behavior across encoder and decoder; CLIP exhibits elevated TE in its text tower but late-layer redundancy in its vision tower; and LLaVA-1.5-7B shows a clear ramp-up of vision→text TE in its upper decoder blocks, indicating that multimodal fusion is localized near the top of the language stack. Building on these empirical findings, our contributions are as follows: at the *layer level*, we show that TE exposes depth-dependent

[1]Anonymous Institution, Anonymous City, Anonymous Region, Anonymous Country. Correspondence to: Anonymous Author <anon.email@domain.com>.

Preliminary work. Under review by the International Conference on Machine Learning (ICML). Do not distribute.

redundancy, enabling structured pruning that reduces computation while preserving accuracy; at the *cross-modal level*, we use TE in LLaVA-1.5-7B to characterize how visual information flows into the language tower, pinpointing the layers where vision→text influence is strongest; and at the *head level*, we combine CLS/EOT attention mass with TE to obtain role-aware head classification, yielding interpretable attribution maps within each layer.

Our experiments highlight the stability and universality of TE profiles across tasks, datasets, and adaptation regimes such as LoRA. TE-based pruning achieves competitive performance on GLUE and MSCOCO-2014 compared to existing baselines, while offering interpretable diagnostics of where redundancy arises. More broadly, TE provides a unified lens for analyzing, interpreting, and compressing large Transformer-based models, contributing to both scientific understanding and practical efficiency in the era of foundation models.

## 2. Related Work

**Information-theoretic analyses in deep learning.** Mutual information (MI) and information–bottleneck (IB) viewpoints have long been used to analyze representation learning and training dynamics in neural networks (Tishby et al., 1999; Shwartz-Ziv & Tishby, 2017; Saxe et al., 2018). Practical MI estimation in high dimensions is challenging; popular approaches include $k$-nearest neighbor estimators (Kraskov et al., 2004), contrastive bounds such as CPC/InfoNCE (van den Oord et al., 2018), and variational estimators like MINE and related lower bounds (Belghazi et al., 2018; Poole et al., 2019). Our work departs from undirected MI and focuses on *directed*, layer-to-layer information flow during training.

**Transfer entropy and Granger causality.** Transfer entropy (TE) was introduced by (Schreiber, 2000) as a non-parametric, model-free measure of directed information transfer between stochastic processes. Extensions include local/pointwise TE (Lizier, 2012), methodological tutorials and neuroscience applications (Vicente et al., 2011; Bossomaier et al., 2016), and connections to Granger causality (GC): under linear–Gaussian assumptions, TE reduces to GC (Barnett et al., 2009). Estimating TE in high dimensions typically relies on $k$NN estimators (Kraskov et al., 2004), Gaussian state–space surrogates (Barnett et al., 2009; Lizier et al., 2008), or variational bounds (Belghazi et al., 2018; Mondal et al., 2020). Related "neural Granger" formulations learn sparse temporal dependencies with deep models (Tank et al., 2021). We adopt TE as a *layerwise*, training-time diagnostic for Transformers and derive a linear–Gaussian approximation that enables scalable estimation from cached activations.

**Representation similarity and layer interaction in Transformers.** Layerwise analyses of Transformers often quantify representation *similarity* rather than directed influence, e.g., Singular Vector Canonical Correlation Analysis (SVCCA) (Raghu et al., 2017), Centered Kernel Alignment (CKA) (Kornblith et al., 2019), and cosine-based stability measures. Attribution- and attention-based tools (attention rollout/flow and gradient-based explanations) probe token-level pathways (Abnar & Zuidema, 2020; Chefer et al., 2021; Sundararajan et al., 2017), but they do not directly yield a measure of *new* information contributed by one layer to the next during learning. Our approach complements these by providing a directed, conditional information measure, $I(X_{\ell+1}^{t+1}; X_\ell^{t+1} \mid X_{\ell+1}^t)$, tailored to layerwise dynamics.

**Efficiency and layer pruning.** Reducing inference cost in LLM/VLM commonly leverages quantization (Frantar et al., 2022; Dettmers et al., 2023), sparse/structured pruning (Frantar & Alistarh, 2023; Ma et al., 2023), and layer dropping or replacement (Fan et al., 2019; Chen et al., 2025; Guo et al., 2025). Many methods rank layers by activation similarity or magnitude-based proxies, which only indirectly reflect a block's functional contribution. Closest to our perspective are functional criteria based on derivatives (e.g., Jacobian properties) used to study stability and sensitivity (Pennington et al., 2017; Yang, 2019). Our contribution is to employ *transfer entropy*—a directed information measure—at the layer level to characterize learning dynamics, and to show how high $\text{TE}_\ell(t)$ can inform pruning policies by identifying upper layers that add little *unique* information beyond their inputs.

**Vision–language models (VLMs).** Modern VLMs align visual and textual representations via contrastive pre-training and/or generative instruction tuning, including CLIP (Radford et al., 2021b), ALIGN (Jia et al., 2021), Flamingo (Alayrac et al., 2022), BLIP/BLIP-2 (Li et al., 2022; 2023), PaLI/PaLI-X (Chen et al., 2022), and LLaVA (Liu et al., 2023). Analyses of VLM internals typically rely on probing, linear evaluation, or similarity metrics to assess cross-modal fusion. In contrast, we propose TE as a *layerwise, directed* measure that captures how updates propagate across adjacent blocks during fine-tuning, offering a complementary view of fusion dynamics and capacity usage.

**Sink/Broadcaster/Benign Classification** Prior work has shown that attention heads in Transformers often specialize into distinct functional roles, such as syntactic tracking or information aggregation (Clark et al., 2019; Voita et al., 2019; Michel et al., 2019). More recent studies extend this perspective to vision and multimodal models, highlighting how certain heads act as strong collectors into the special token (sinks), while others distribute information outward

(broadcasters), leaving a large fraction of benign heads with limited directional influence (Raghu et al., 2021; Radford et al., 2021a). This line of research builds on broader interpretability debates around whether attention is explanatory (Jain & Wallace, 2019; Serrano & Smith, 2019) and complements flow- and attribution-based analyses (Abnar & Zuidema, 2020; Chefer et al., 2021).

## 3. Introduction to Transfer Entropy

Transfer entropy (TE), originally introduced by Schreiber (Schreiber, 2000), is an information-theoretic measure that quantifies the *directed* flow of information between two stochastic processes. Unlike symmetric measures such as mutual information, TE captures the *asymmetric* influence of one process on another, making it a powerful tool for identifying directional dependencies and potential causal relationships.

In the context of neural network training, and in particular large language models (LLMs), TE naturally extends to measuring how the representation at one layer influences the representation at another layer in the *next training step*, after controlling for the layer's own prior state. This enables us to track how updates in a lower layer propagate upwards in the network over time, isolating the component of change attributable to inter-layer communication rather than internal memory.

For two discrete-time stochastic processes $X(t)$ and $Y(t)$, the transfer entropy from $X$ to $Y$ is defined as (Schreiber, 2000):

$$T_{X \to Y} = \sum_t p(y_{t+1}, y_t, x_t) \, \log \frac{p(y_{t+1} \mid y_t, x_t)}{p(y_{t+1} \mid y_t)}, \quad (1)$$

where $p(\cdot)$ denotes the joint or conditional probability mass/density functions of the relevant variables.

Equivalently, TE can be expressed as a conditional mutual information (CMI) (Shahsavari Baboukani et al., 2020):

$$T_{X \to Y} = I(Y_{t+1}; X_t \mid Y_t), \quad (2)$$

where $I(\cdot; \cdot \mid \cdot)$ is the CMI between $Y_{t+1}$ and $X_t$ given $Y_t$. This form reveals that TE measures the additional predictive information about the *future* of $Y$ contributed by the *present* of $X$, after removing the influence of $Y$'s own present state.

Because TE is inherently time-directed and conditioned on the receiver's state, it is well suited to studying training dynamics in deep networks, where representations evolve step by step. In what follows, we instantiate TE at the level of adjacent layers during fine-tuning of large language models (LLMs), turning it into a per-step, layer-to-layer influence measure.

## 4. Transfer Entropy for Layerwise Information Flow

### 4.1. Problem Formulation

In our LLM/VLM setting, we interpret the two processes $X(t)$ and $Y(t)$ in TE as the activations of layer $l$ and layer $l+1$, respectively, evolving over training. The corresponding instantaneous transfer entropy is

$$\text{TE}_l(t) = I(X_{l+1}^{t+1}; X_l^t \mid X_{l+1}^t), \quad (3)$$

which isolates the influence of updated inputs from layer $l$ on the updated outputs of layer $l+1$, beyond what can be explained by the latter's own previous state. Tracking $\text{TE}_l(t)$ across training steps yields a dynamic, directed view of inter-layer information flow during fine-tuning.

Computing $\text{TE}_l(t)$ exactly is intractable in LLMs and VLMs. The definition in (3) requires the true joint and conditional probability densities $p(X_{l+1}^{t+1}, X_l^t, X_{l+1}^t)$, $p(X_{l+1}^t, X_{l+1}^t)$, and $p(X_{l+1}^{t+1} \mid X_{l+1}^t)$, which are unknown for high-dimensional, continuous-valued layer activations. Estimating these densities without strong assumptions is impractical due to the curse of dimensionality: hidden states in modern LLMs and VLMs often have thousands of dimensions, far exceeding what can be reliably covered by finite probe batches. Moreover, the dependencies between $X_{l+1}^{t+1}$, $X_l^{t+1}$, and $X_{l+1}^t$) are highly nonlinear, further complicating density estimation. As a result, practical TE computation relies on approximations, such as $k$-nearest neighbor estimators (Kraskov et al., 2004), Gaussian models (Barnett et al., 2009)(Lizier et al., 2008), or variational mutual information bounds (Belghazi et al., 2018)(Mondal et al., 2020), which yield tractable, sample-based estimates but cannot recover the exact TE value.

### 4.2. Approximation of TE

**Local perturbation model and Jacobians.** Fix a layer index $l$ and two consecutive training steps $t$ and $t+1$. We study the directed dependence from the state of layer $l$ at step $t$ to the state of layer $l+1$ at step $t+1$, conditioned on the previous state of layer $l+1$ at step $t$. Around the operating point $x_0 = (X_l^t, X_{l+1}^t)$, inject a small, zero-mean Gaussian perturbation $\delta x \sim \mathcal{N}(0, \sigma^2 I)$ that probes the local response of adjacent layers. The details on the role of the perturbation $\delta x$ is provided in Appendix A. Let

$$J_l = \left. \frac{\partial X_l^{t+1}}{\partial x} \right|_{x=x_0}, \qquad J_{l+1} = \left. \frac{\partial X_{l+1}^{t+1}}{\partial x} \right|_{x=x_0}, \quad (4)$$

and introduce the shorthand random variables

$$u = X_{l+1}^{t+1}, \qquad y = X_{l+1}^t, \qquad v = X_l^{t+1}. \quad (5)$$

A first-order Taylor expansion yields the local linear responses

$$u \approx u_0 + J_{l+1}\,\delta x, \tag{6}$$
$$v \approx v_0 + J_l\,\delta x, \tag{7}$$
$$y \approx y_0, \tag{8}$$

where $(u_0, v_0, y_0)$ are deterministic offsets at $x_0$ and do not affect covariances. Crucially, $y$ is evaluated at time $t$ and thus carries no $\delta x$–induced variability at this local scale.

**Theorem 4.1** (Jacobian Linear–Gaussian Approximation of Transfer Entropy (with $v$)). *Let $z := (v, y)$ and define the covariance blocks induced by (6)-(8):*

$$\Sigma_{uu} = \sigma^2\,J_{l+1}J_{l+1}^\top, \tag{9}$$
$$\Sigma_{uy} = 0, \tag{10}$$
$$\Sigma_{uz} = \sigma^2\,J_{l+1}\begin{pmatrix} J_l \\ J_{l+1} \end{pmatrix}^\top, \tag{11}$$
$$\Sigma_{zz} = \sigma^2\begin{pmatrix} J_lJ_l^\top & J_lJ_{l+1}^\top \\ J_{l+1}J_l^\top & J_{l+1}J_{l+1}^\top \end{pmatrix}. \tag{12}$$

*Define the positive semidefinite matrix*

$$A(z) = \Sigma_{uu}^{-1/2}\,\Sigma_{uz}\,\Sigma_{zz}^{-1}\,\Sigma_{zu}\,\Sigma_{uu}^{-1/2}. \tag{13}$$

*Then the layerwise transfer entropy admits the determinant form*

$$\mathrm{I}\big(X_{l+1}^{t+1};\, X_l^{t+1} \mid X_{l+1}^t\big) \approx -\frac{1}{2}\,\ln\det\big(I - A(z)\big), \tag{14}$$

*and, when $\|A(z)\|$ is small, the first-order trace approximation*

$$\mathrm{I}\big(X_{l+1}^{t+1};\, X_l^{t+1} \mid X_{l+1}^t\big) \approx \frac{1}{2}\,\mathrm{tr}\big(A(z)\big). \tag{15}$$

The proof of this Theorem 4.1 is provided in Appendix B.

*Remark* 4.2 (Interpretation and edge cases). If $J_l = 0$ (no local sensitivity of $v$ to the probe), then $A(z)$ collapses and $\mathrm{I}(X_{l+1}^{t+1}; X_l^{t+1} \mid X_{l+1}^t) \approx 0$. If $J_l$ aligns strongly with $J_{l+1}$, observing $v$ explains a large portion of the variability in $u$ beyond $y$, and TE increases accordingly.

**Theorem 4.3** (Cosine–Similarity Approximation of Transfer Entropy (with $v$)). *Let a mini-batch contain embeddings $\{\mathbf{z}_{l,i}, \mathbf{z}_{l+1,i}\}_{i=1}^B$ for consecutive samples (or tokens) indexed by $i$. Form finite differences*

$$\Delta\mathbf{z}_{l,i} = \mathbf{z}_{l,i+1} - \mathbf{z}_{l,i}, \tag{16}$$
$$\Delta\mathbf{z}_{l+1,i} = \mathbf{z}_{l+1,i+1} - \mathbf{z}_{l+1,i}, \tag{17}$$

*which serve as surrogates for $J_l\delta x_i$ and $J_{l+1}\delta x_i$, respectively. Under a rank-one covariance approximation for each $i$ (augmented with a small ridge $\epsilon I$), the Jacobian linear–Gaussian TE in (15) admits the practical estimator*

$$\mathrm{I}\big(X_{l+1}^{t+1};\, X_l^{t+1} \mid X_{l+1}^t\big) \approx \frac{1}{2(B-1)} \sum_{i=1}^{B-1} \cos^2\big(\Delta\mathbf{z}_{l+1,i},\, \Delta\mathbf{z}_{l,i}\big) \tag{18}$$

*and averaging (18) over batches yields a stable epoch-level estimate.*

The proof of Theorem 4.3 is provided in Appendix C.

**TE versus Granger Causality.** Transfer entropy is closely related to Granger causality, but the two notions arise from different formalisms. Granger causality defines a directed influence from a source process $X$ to a target process $Y$ if past values of $X$ improve the prediction of $Y$ beyond what is achievable from the past of $Y$ alone, typically operationalized via linear autoregressive models and variance-based hypothesis tests. In contrast, TE quantifies the same notion of directed influence using conditional mutual information in (3), and is therefore model-agnostic and, in principle, applicable to nonlinear and non-Gaussian dependencies. In the linear–Gaussian setting, the two measures are mathematically equivalent: Granger causality reduces to a log-ratio of prediction-error variances, while TE reduces to the same quantity up to a constant factor, as shown in (Barnett et al., 2009). Our Jacobian-based linear–Gaussian approximation in Theorem 4.1 can thus be viewed as a multivariate, local Granger-style model expressed in information-theoretic terms. However, our cosine-similarity estimator in Theorem 4.3 avoids fitting explicit autoregressive models and directly operates on high-dimensional layer activations in LLMs and VLMs, making it more practical in the large-scale neural setting.

Appendix D discusses the validity and error control for our TE surrogates. In Appendix E, we summarize the procedure for computing instantaneous, layerwise transfer entropy using the cosine-similarity approximation (Algorithm 1). Appendix F compares the computational complexities of exact TE and its approximations. To illustrate their closeness in practice, an illustrative example is provided in Appendix G.

### 4.3. TE for Layer Pruning in LLM/VLM

**Rationale.** The layerwise instantaneous TE in (3) quantifies how much of the *new* information in the updated upper-layer state $X_{\ell+1}^{t+1}$ is attributable to the updated lower-layer state $X_\ell^{t+1}$, beyond what was predictable from the upper layer's own prior $X_{\ell+1}^t$. When $\mathrm{TE}_\ell(t)$ is large, the update at layer $\ell$ already explains most of the change at layer $\ell+1$; equivalently, layer $\ell+1$ contributes little *unique* information. This suggests a pruning rule that targets layer $\ell+1$ when the corresponding $\mathrm{TE}_\ell$ is persistently high.

**Score and selection.** Define a smoothed, data-aware pruning score for the $(\ell+1)$-th layer by temporally averaging (or Exponential Moving Average (EMA) smoothing) the instantaneous TE estimates from Algorithm 1. Let $\mathcal{T}$ denote the set of training steps over which TE is aggregated (e.g., one epoch or a sliding window), $S_{\ell+1}^{\text{prev}}$ the score from the previous iteration in the EMA update, and $S_{\ell+1}^{\text{null}}$ a baseline score computed from a *null* TE estimator in which $X_l^{t+1}$ is randomly permuted within a batch to break the $(\ell \to \ell+1)$ dependency.

The smoothed pruning score is given by

$$S_{\ell+1} = \frac{1}{|\mathcal{T}|} \sum_{t \in \mathcal{T}} \widehat{\text{TE}}_\ell(t) \qquad (19)$$

or

$$S_{\ell+1} = (1-\alpha)\, S_{\ell+1}^{\text{prev}} + \alpha\, \widehat{\text{TE}}_\ell(t), \qquad (20)$$

where $\alpha \in (0,1)$ is the EMA smoothing coefficient.

To improve robustness, we calibrate the score by subtracting the null baseline:

$$\widetilde{S}_{\ell+1} = \max\left\{0,\ S_{\ell+1} - \lambda\, S_{\ell+1}^{\text{null}}\right\}, \qquad \lambda \in [0,1], \quad (21)$$

which suppresses spurious TE signals that may arise from stochasticity or batch-level correlations.

Given a pruning budget $K$ (or a threshold $\tau$), we then rank layers by $\widetilde{S}_{\ell+1}$ and prune the top-$K$ layers (or all layers satisfying $\widetilde{S}_{\ell+1} \geq \tau$).

**Policy and guardrails.** We apply the rule "*higher* $\text{TE}_\ell$ $\Rightarrow$ *prune layer* $\ell+1$" subject to structural constraints typical in LLM/VLM: do not prune embeddings, the first fusion (cross-attention) block, or the final prediction head; enforce a minimum depth per module (vision encoder, text encoder, fusion decoder). In pre-LN Transformers, pruning layer $\ell+1$ is implemented by replacing its block with an identity (skip) mapping of the same shape, preserving residual pathways and masks. A short post-pruning fine-tune stabilizes the network.

# 5. Experimental Results

## 5.1. Information Flow in LLMs

Our experiments on information flow span three major Transformer families, represented by four widely used architectures: RoBERTa (Liu et al., 2019) as an *encoder-only* model, T5 (Raffel et al., 2020b) as a full *encoder–decoder* sequence-to-sequence model, and Llama-3.2-3B (Dubey et al., 2024) and Qwen-2.5-7B (Qwen Team, 2024) as *decoder-only* autoregressive LLMs. This selection enables us to analyze how TE behaves across fundamentally different architectural paradigms.

### 5.1.1. TE IN FULL FINE TUNING

We evaluate TE of RoBERTa under full fine tuning using on the GLUE benchmark (Wang et al., 2018). In Fig. 1, we summarize the TE versus the layer index (starting layer 2 to compute TE) in RoBERTa$_{\text{base}}$ fine tuning for eight datasets on the GLUE benchmark. Across all datasets, the TE as a function of layer depth exhibits a highly consistent and distinctive profile. Specifically, the TE remains low in the input and output layers, but increases sharply through the early and middle layers, reaching a pronounced peak in the upper-middle layers of the model (typically layers 3 to 8). This non-monotonic "∩-shaped" pattern is robust to both the choice of dataset and the epoch of fine-tuning. The peak TE values, observed in the middle layers, often surpass 0.2–0.3, while the TE in the final layer can be as low as $10^{-3}$. Minor variations in the exact location and amplitude of the TE peak are observed across different datasets, reflecting some degree of task specificity, yet the qualitative trend remains universal. Furthermore, this profile is stable across training epochs, with early, middle, and late layers maintaining their respective TE characteristics throughout fine-tuning.

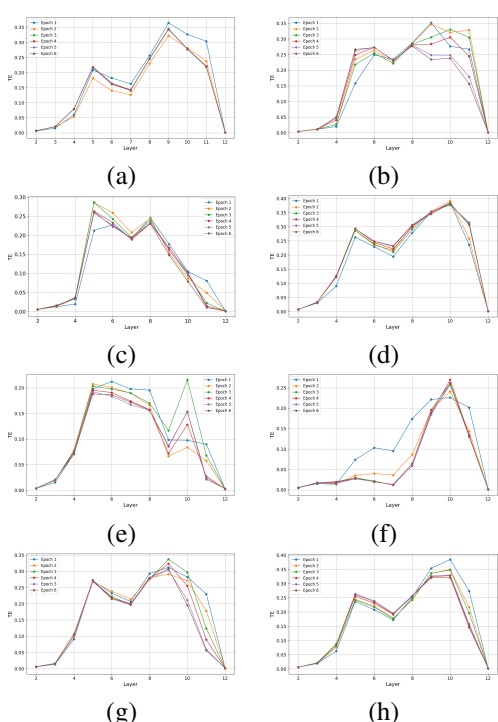

*Figure 1.* The TE versus layer index in RoBERTa$_{\text{base}}$ with full fine tuning for 6 epochs using eight datasets on the GLUE benchmark, (a) MNLI, (b) MRPC, (c) SST-2, (d) CoLA, (e) QNLI, (f) QQP, (g) RTE, (h) STS-B.

These results indicate a clear functional specialization across the layers of large language models. The low TE in the input layers suggests stable feature extraction, while the elevated TE in the middle layers marks a zone of increased represen-

tational uncertainty and flexibility, likely corresponding to dynamic knowledge integration and task adaptation. Output layers, in contrast, return to low TE, reflecting highly deterministic, task-aligned decoding. The universality of this ∩-shaped TE profile implies that TE is an intrinsic property of transformer architectures during fine-tuning, largely independent of specific dataset or task.

Consequently, TE can serve as a principled metric for identifying which layers are most amenable to pruning or compression. Layers with high TE exhibit greater redundancy in their representations, often reflecting dynamic adaptation rather than essential information. As a result, pruning these layers with higher TE is less likely to harm the model's predictive accuracy or core capabilities, since their information is either noisy or can be compensated by neighboring layers. In contrast, low TE layers encode more stable and indispensable representations. This observation justifies a TE-guided pruning strategy, where layers with the highest TE are removed to achieve efficient model compression without significant degradation in performance.

### 5.1.2. APPLICATION TO LLM PRUNING

We prune layers using the TE. Intuitively, TE quantifies how much a layer's representation change is predictable from (or duplicative of) neighboring layers, i.e., how little novel, task-relevant information it contributes. Layers with higher TE thus encode more redundant or noisy variation and are stronger pruning candidates. We therefore rank layers by TE (higher is more prunable) and remove those at the top of the ranking subject to architectural constraints (e.g., preserving embeddings and the output head). This targeted criterion preferentially eliminates duplicative computation, reducing model size and inference latency while maintaining—often after a brief recovery fine-tune—accuracy and generalization.

Table 1 reports pruning results on RoBERTa$_{base}$ across the GLUE benchmark (MNLI, MRPC, SST-2, CoLA, QNLI, QQP, RTE, STS-B). The first row gives the fully fine-tuned baseline after 6 epochs, using the task-appropriate metric (accuracy for MNLI/QNLI/RTE/SST-2, Matthews correlation for CoLA, F1/accuracy for MRPC/QQP, and Pearson correlation for STS-B). For pruning, we remove four of the twelve transformer layers using four criteria: (1) pruning the four layers with the highest TE, (2) pruning the four layers with the highest knowledge entropy (KE) (Kim et al., 2025), (3) random structured LayerDrop following Fan et al. (Fan et al., 2019), and (4) SlimLLM layer pruning (Guo et al., 2025). After each pruning/dropout operation, we fine-tune for 5 additional epochs. The post-pruning results are summarized in rows 2–5.

To assess whether TE provides a meaningful pruning signal, we also conduct a *reverse-TE pruning* experiment in which

we prune the four layers with the *lowest* TE rather than the highest. Row 6 of Table 1 presents the corresponding results. As expected, pruning low-TE layers leads to substantial degradation, indicating that these layers introduce information that cannot be predicted from their predecessors. Conversely, high-TE layers are more redundant.

Our results show that TE pruning closely tracks the full fine-tuning baseline, indicating that TE is effective at identifying layers that contribute little novel, task-relevant information. In contrast, KE-based pruning causes substantial degradation across all tasks, suggesting it is not a reliable signal for structured layer removal in RoBERTa. LayerDrop yields intermediate performance—better than KE but consistently below the TE criterion, with especially large gaps on linguistically challenging tasks such as CoLA and QNLI. SlimLLM is competitive but generally trails our method and the no-pruning baseline, with parity on QNLI and near-parity on QQP/STS-B. Overall, these findings support the TE as a robust and principled pruning signal, enabling meaningful compression with minimal loss in downstream performance.

### 5.1.3. ADDITIONAL EXPERIMENTS ON INFORMATION FLOW IN LLMS

Appendix H.1 compares TE with knowledge entropy (KE) (Kim et al., 2025) under full fine-tuning; Appendix H.2 evaluates TE under LoRA (Hu et al., 2022); and Appendix H.3 studies learning-rate effects. We further analyze TE and TE-guided pruning in T5 on e-SNLI, ANLI, CommonsenseQA, and SVAMP (Appendix H.4–H.5), and test scalability on Llama-3.2-3B and Qwen-2.5-7B (GLUE + LoRA) in Appendix H.6. Overall, TE provides a stable, complementary view to KE and supports compression. In RoBERTa, KE decreases with depth while TE peaks mid-layer; LoRA leaves TE largely invariant, whereas overly large learning rates collapse TE and accuracy. T5 shows distinct encoder/decoder TE shapes, and TE-guided pruning matches or exceeds full fine-tuning on CoT tasks. Across 12 datasets, mean TE reflects architecture while TE variance captures dataset-driven plasticity. For larger decoder-only models, TE is ∩-shaped: low early, high across a mid-depth compute band, then sharply lower in final layers.

### 5.2. Information Flow in VLMs

#### 5.2.1. LAYERWISE TE PROFILES AND PRUNING IN CLIP ViT-B/16

We apply LoRA (Hu et al., 2022) fine-tuning to CLIP ViT-B/16 (Radford et al., 2021b), whose vision and text encoders each consist of 12 transformer layers. LoRA fine-tuning ($r = 8$) is performed for two epochs on the MSCOCO2014 dataset (Lin et al., 2014). Figure 2 plots TE$_\ell$ against layer index $\ell$, where TE$_\ell$ denotes the transfer-entropy proxy between layer $\ell$ and its successor $\ell+1$ within either the vision

*Table 1.* GLUE benchmark accuracy (or correlation for STS-B) for RoBERTa$_{\text{base}}$ under full fine-tuning, TE pruning, KE pruning, layer dropout, and SlimLLM.

| Method | MNLI | MRPC | SST-2 | CoLA | QNLI | QQP | RTE | STS-B |
|---|---|---|---|---|---|---|---|---|
| No Pruning | 0.7840 | **0.8480** | **0.9220** | **0.8274** | **0.8731** | **0.8386** | **0.7509** | **0.9075** |
| Our TE Pruning | **0.7860** | **0.8480** | 0.9186 | 0.8255 | 0.8548 | 0.8361 | 0.7437 | 0.9063 |
| KE Pruning | 0.3668 | 0.6838 | 0.7294 | 0.6913 | 0.5054 | 0.6318 | 0.5271 | 0.3125 |
| Layer Dropout | 0.6910 | 0.8235 | 0.9186 | 0.6913 | 0.8371 | 0.8003 | 0.7256 | 0.8938 |
| SlimLLM | 0.7660 | 0.8088 | 0.8979 | 0.8092 | 0.8548 | 0.8359 | 0.6787 | 0.9043 |
| Low-TE Pruning | 0.3470 | 0.6838 | 0.9094 | 0.6913 | 0.5094 | 0.6510 | 0.4729 | 0.2896 |

or text stack.

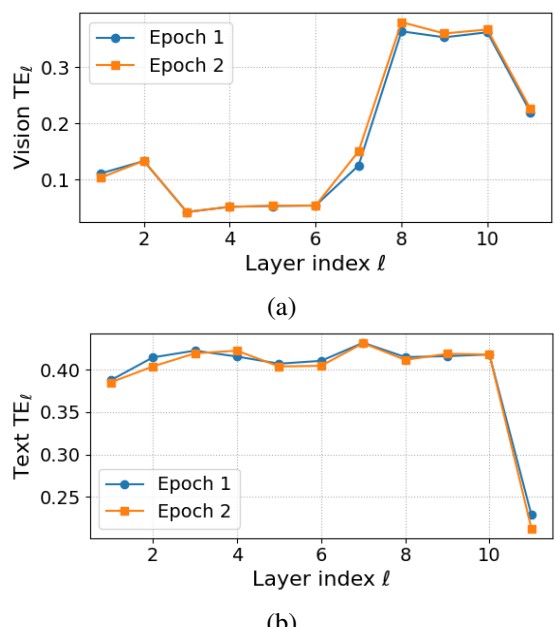

(a)

(b)

*Figure 2.* Layerwise TE profile of CLIP ViT-B/16 in LoRA ($r = 8$) fine tuning. (a) Vision TE$_\ell$. (b) Text TE$_\ell$.

As shown in Fig. 2a, across the 12-block ViT-B/16 vision backbone, the TE profile exhibits a clear depth-dependent structure. Early-to-mid transitions ($\ell=1$–7) yield consistently low TE$_\ell$, indicating that their successors are non-redundant and perform distinct transformations. By contrast, late transitions ($\ell=8$–10) peak sharply, while the terminal transition ($\ell=11$) decreases again. Under the successor-pruning rule, redundancy therefore concentrates in the successors of $\ell=8$–10, identifying layers 9–11 as primary pruning targets. This pattern highlights a hierarchical importance: beginning–mid layers are essential for representation building, whereas latter layers are comparatively compressive and thus more easily removed. This observation aligns with recent findings that CLIP exhibits a hierarchical organization similar to that of the human brain (Yang et al., 2024). In this view, early network layers correspond to early brain regions and are responsible for learning simpler concepts,

while later layers correspond to higher-level brain regions and capture increasingly complex concepts. Such hierarchical alignment has been shown to facilitate fine-tuning on small datasets without catastrophic forgetting. In Fig. 2b, the text encoder displays a different behavior. The TE curve remains uniformly high across most of the depth ($\ell=1$–10), with only a noticeable dip at the final transition ($\ell=11$). Consequently, many successors (2–11) emerge as redundancy candidates, while the last layer (successor of $\ell=11$) should be preserved. Because high TE$_\ell$ values are broadly distributed rather than localized, pruning must be applied selectively and with spacing (e.g., choosing non-adjacent successors at prominent peaks) to avoid excessive loss of depth.

After LoRA (with rank 8) warm-up for two epochs, we applied TE-guided layer pruning followed by three epochs of LoRA fine-tuning. We also compared our approach with Short-LVLM (Ma et al., 2025) and streamline pruning (Chen et al., 2025). The results are summarized in Table 2, which demonstrates that TE-based pruning consistently outperforms the other two approaches under matched pruning budgets. This indicates that TE avoids pruning functionally critical layers more effectively. Across all settings, text-to-image recall decreases more sharply than image-to-text, consistent with the known asymmetry of CLIP retrieval.

### 5.2.2. ADDITIONAL EXPERIMENTS ON INFORMATION FLOW IN VLMs

In Appendix I.1, we analyze how LLaVA-1.5-7B (Liu et al., 2023)(Liu et al., 2024) processes and propagates visual and text information. Across both Flickr8k and MSCOCO, we observe a highly consistent hierarchy of information flow in LLaVA-1.5-7B. The CLIP ViT-L/14 vision tower shows a stable, dataset-invariant TE pattern with peaks in mid-to-late layers, reflecting the progression from low-level patches to higher-level semantics; LoRA updates do not affect this frozen backbone. The LLaMA-7B language tower exhibits uniformly high TE ($\sim$0.30–0.50), consistent with deep autoregressive refinement, with only mild shifts in magnitude across epochs. The most distinctive trend is the steadily increasing cross-modal TE from lower to higher layers: early

Table 2. Retrieval performance (MSCOCO) of CLIP ViT-B/16 under different pruning strategies.

| Method | Pruned Layers | Image-to-Text Recall (%) | | | Text-to-Image Recall (%) | | |
|---|---|---|---|---|---|---|---|
| | | R@1 | R@5 | R@10 | R@1 | R@5 | R@10 |
| No Pruning | – | 42.20 | 66.21 | 74.94 | 26.66 | 48.94 | 58.91 |
| Our TE pruning | 2V+2T | 27.36 | 51.44 | 62.78 | 16.67 | 36.50 | 47.03 |
| | 4V+4T | 15.95 | 34.81 | 45.51 | 9.53 | 24.08 | 33.11 |
| | 5V+5T | 11.79 | 28.42 | 38.68 | 7.07 | 19.49 | 27.92 |
| Short-LVLM | 2V+2T | 25.65 | 49.34 | 60.63 | 15.91 | 35.16 | 45.65 |
| | 4V+4T | 13.10 | 30.33 | 40.28 | 7.95 | 20.85 | 29.29 |
| | 5V+5T | 8.02 | 20.62 | 28.81 | 4.87 | 14.19 | 20.91 |
| Streamline pruning | 2V+2T | 26.83 | 50.26 | 60.83 | 16.18 | 35.38 | 45.66 |
| | 4V+4T | 12.02 | 28.49 | 38.18 | 7.57 | 20.03 | 28.37 |
| | 5V+5T | 9.34 | 23.07 | 31.63 | 5.67 | 16.15 | 23.54 |

decoder blocks show minimal visual influence, while upper layers absorb and propagate most of the visual perturbation. This upward vision→text flow is present in both datasets and becomes sharper with training, especially under smaller data. Using TE, we further analyze information flow at the head levels in CLIP. At the head level (Appendix I.2), combining TE with CLS/EOT attention mass enables role-aware head classification, yielding interpretable maps of functional responsibility across depth. We identify per-layer head roles driven jointly by CLS/EOT attention mass and TE impact. Vision heads exhibit a mixed, localized pattern, while text heads display stronger depth-wise polarization.

# 6. Conclusions and Future Work

We introduced a TE perspective on information flow in Transformers and instantiated it with practical, perturbation–based estimators that operate at the granularity of *layers* and *attention heads*. The resulting diagnostics cleanly separate *where probability mass goes* (via attention) from *which modules actually drive representational updates* (via TE), yielding interpretable maps of responsibility across both depth and modality. Empirically, TE profiles reveal consistent depth-dependent structures: 1) in encoder–only LLMs such as RoBERTa, TE rises to a pronounced peak in the middle layers and diminishes at the input and output extremes; 2) for decoder-only LLMs such as Llama-3.2-3B and Qwen-2.5-7B, TE consistently highlights a robust band of mid-level layers responsible for most task-dependent computation, whereas the earliest and final layers act largely as input/output adapters; 3) in T5, encoder/decoder dynamics diverge, with the encoder showing stable high TE and the decoder peaking mid-depth; 4) in CLIP ViT-B/16, redundancy concentrates late in the vision tower while the text tower maintains a broad and elevated TE signature; 5) in LLaVA-1.5-7B, the vision→text TE ramps up sharply in the

upper decoder blocks, indicating that multimodal fusion is localized near the top of the language stack. These profiles are robust under LoRA fine-tuning, predictive of pruning tolerance, and sensitive to optimization regimes—for example, collapsing under overly large learning rates. Using TE as a pruning criterion, we demonstrated that models can be compressed while preserving task accuracy in LLMs and achieving competitive retrieval in CLIP, outperforming alternative heuristics under equivalent pruning budgets.

Looking ahead, several promising research directions emerge. (i) *Online usage:* integrate TE as a streaming diagnostic during training to enable adaptive depth, dynamic early exiting, and optimizer or curriculum scheduling. (ii) *Role-aware systems:* leverage sink/broadcaster maps to guide routing decisions, Key Value-cache allocation, quantization strategies, and targeted placement of adapters or LoRA modules. (iii) *Cross-attention and multi-modal flow:* extend TE to encoder–decoder interactions and cross-tower fusion blocks in VLMs, providing a principled measure of inter-modal communication. (iv) *Robustness and safety:* apply TE to identify brittle or spurious broadcasters, improve interpretability via attribution without gradients, and develop defenses against failure modes arising from pathological information flow.

Taken together, TE provides a unifying, data-driven lens on the dynamics of representation change in large models. By complementing similarity-based diagnostics with a *directed*, conditional measure of information transfer, TE supports both deeper scientific insight and practical interventions for compression, interpretability, and safety. The limitations of our TE approximations are discussed in Appendix K. We believe that principled, scalable TE proxies will increasingly become part of the standard toolkit for diagnosing, compressing, and steering large foundation models across domains.

## Impact Statement

This paper presents work whose goal is to advance the understanding of information flow in large language models and vision-language models. The methods introduced are analytical and diagnostic in nature, and do not introduce new model capabilities or applications. The potential societal and ethical implications of this work are those generally associated with advances in machine learning interpretability and model efficiency, none of which we believe require specific discussion beyond established considerations.

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

## A. Understanding the Perturbation

This section clarifies the role of the perturbation $\delta x$ used in the Jacobian linear–Gaussian approximation of transfer entropy in Section 4.2. Importantly, the perturbation is *not* assumed to generate an additive temporal update of the form

$$\left(X_l^{t+1}, X_{l+1}^{t+1}\right) \approx \left(X_l^t, X_{l+1}^t\right) + \delta x. \tag{22}$$

Such an assumption would collapse the Jacobians to the identity and render TE trivial. Instead, $\delta x$ acts as a *local probe* for the transition map between steps $t$ and $t+1$.

**Perturbations as probes of the transition map.** We model the update from step $t$ to step $t+1$ as a deterministic map

$$F: \ x = (X_l^t, X_{l+1}^t) \ \longmapsto \ \left(X_l^{t+1}(x), X_{l+1}^{t+1}(x)\right). \tag{23}$$

The perturbation $\delta x \sim \mathcal{N}(0, \sigma^2 I)$ is applied in the *input space of $F$* around an operating point $x_0$. A first–order Taylor expansion yields

$$X_l^{t+1}(x_0 + \delta x) \approx v_0 + J_l \, \delta x, \tag{24}$$

$$X_{l+1}^{t+1}(x_0 + \delta x) \approx u_0 + J_{l+1} \, \delta x, \tag{25}$$

where $J_l$ and $J_{l+1}$ capture how infinitesimal changes to $(X_l^t, X_{l+1}^t)$ propagate through the transition. These Jacobians encode the local sensitivity structure that underlies the TE approximation.

**Conditioning on $X_{l+1}^t$ and the role of $y_0$.** The layerwise TE we approximate is

$$\mathrm{TE}_l(t) = I\left(X_{l+1}^{t+1}; X_l^{t+1} \,\middle|\, X_{l+1}^t\right), \tag{26}$$

so $X_{l+1}^t$ serves as the conditioning variable. In the Jacobian approximation, however, we work *locally* around a fixed realization of the past state: for a given context $y_0$, we consider the TE

$$I\left(u; v \,\middle|\, y = y_0\right) = I\left(X_{l+1}^{t+1}; X_l^{t+1} \,\middle|\, X_{l+1}^t = y_0\right), \tag{27}$$

and approximate this quantity by linearizing the transition map at the operating point $x_0$ with $X_{l+1}^t = y_0$. In this local view, $y$ is treated as a fixed context and only the innovation $\delta x$ induces variability in $(u, v)$. Concretely, the linearized model around $(x_0, y_0)$ takes the form

$$u \approx u_0 + J_{l+1} \, \delta x, \tag{28}$$

$$v \approx v_0 + J_l \, \delta x, \tag{29}$$

$$y \approx y_0, \qquad \text{(held fixed)}. \tag{30}$$

The equality $X_{l+1}^t = y_0$ here should therefore be read as: *we are evaluating the conditional TE at a fixed realized value of the past state*, i.e., on the event $\{X_{l+1}^t = y_0\}$. Under this conditioning, $y$ no longer carries any $\delta x$–induced variability, so the covariance between $u$ and $y$ in the local model satisfies

$$\Sigma_{uy} = 0, \tag{31}$$

as used in Theorem 4.1. This corresponds to computing the conditional covariance $\mathrm{Cov}(u, v \mid y = y_0)$ induced by local perturbations of the transition while keeping the past state $X_{l+1}^t$ fixed.

## B. Proof of Theorem 4.1

**Joint Gaussian structure.** Because $u$ and $v$ are affine in the Gaussian perturbation $\delta x$, the pair $(u, z)$ is jointly Gaussian with covariances given by (9)(12), and $y$ is independent of $\delta x$ to first order, hence

$$\Sigma_{uy} = 0 \tag{32}$$

**Conditional entropy** $H(u \mid y)$**.** Since $\Sigma_{uy} = 0$, conditioning on $y$ leaves the covariance of $u$ unchanged at $\Sigma_{uu}$. Therefore

$$H(u \mid y) = \frac{1}{2} \ln\left((2\pi e)^{d_l+1} \det \Sigma_{uu}\right). \tag{33}$$

Intuitively, the local probe $\delta x$ injects variability only through $J_{l+1}$, which $y$ cannot reduce at this scale.

**Conditional entropy** $H(u \mid v, y)$**.** For jointly Gaussian $(u, z)$,

$$\Sigma_{u|z} = \Sigma_{uu} - \Sigma_{uz}\Sigma_{zz}^{-1}\Sigma_{zu}. \tag{34}$$

Factoring out $\Sigma_{uu}^{1/2}$ yields the normalized Schur complement

$$\Sigma_{u|z} = \Sigma_{uu}^{1/2}\left(I - \Sigma_{uu}^{-1/2}\Sigma_{uz}\Sigma_{zz}^{-1}\Sigma_{zu}\Sigma_{uu}^{-1/2}\right)\Sigma_{uu}^{1/2} = \Sigma_{uu}^{1/2}\left(I - A(z)\right)\Sigma_{uu}^{1/2}. \tag{35}$$

Consequently,

$$H(u \mid v, y) = \frac{1}{2} \ln\left((2\pi e)^{d_l+1} \det \Sigma_{uu} \cdot \det\left(I - A(z)\right)\right). \tag{36}$$

**Subtracting and approximating.** Starting from the Gaussian form of conditional entropy,

$$H(u \mid y) = \tfrac{1}{2} \ln\left((2\pi e)^{d_u} \det \Sigma_{u|y}\right), \qquad H(u \mid v, y) = \tfrac{1}{2} \ln\left((2\pi e)^{d_u} \det \Sigma_{u|v,y}\right), \tag{37}$$

subtracting $H(u \mid v, y)$ from $H(u \mid y)$ cancels the $(2\pi e)^{d_u}$ factor and leaves

$$I\left(X_{l+1}^{t+1}; X_l^{t+1} \mid X_{l+1}^t\right) = \tfrac{1}{2} \ln \frac{\det \Sigma_{u|y}}{\det \Sigma_{u|v,y}}. \tag{38}$$

By definition of $A(z)$, we can write

$$\Sigma_{u|v,y} = \left(I - A(z)\right)\Sigma_{u|y}, \tag{39}$$

so the determinants factor as

$$\begin{aligned}
\frac{\det \Sigma_{u|y}}{\det \Sigma_{u|v,y}} &= \frac{\det \Sigma_{u|y}}{\det[(I - A(z))\Sigma_{u|y}]} \\
&= \frac{\det \Sigma_{u|y}}{\det(I - A(z))\,\det \Sigma_{u|y}} \\
&= \frac{1}{\det(I - A(z))}. 
\end{aligned} \tag{40}$$

Thus,

$$I\left(X_{l+1}^{t+1}; X_l^{t+1} \mid X_{l+1}^t\right) = -\tfrac{1}{2} \ln \det(I - A(z)), \tag{41}$$

which is (14). Since $A(z) \succeq 0$ (positive semidefinite), all its eigenvalues $\lambda_i$ satisfy $0 \le \lambda_i \le 1$, hence $I - A(z)$ has eigenvalues in $(0, 1]$, ensuring $\det(I - A(z)) \in (0, 1]$ and making (14) nonnegative, as required for TE.

For the small-$A(z)$ regime, meaning that $\|A(z)\| \ll 1$ in operator norm, the matrix $A(z)$ acts as a small perturbation of the identity in $I - A(z)$. In this case, we can exploit the identity

$$\ln \det M = \operatorname{tr} \ln M, \tag{42}$$

which holds for any positive-definite matrix $M$. Applying this to $M = I - A(z)$ gives

$$\ln \det(I - A(z)) = \operatorname{tr} \ln(I - A(z)). \tag{43}$$

We then expand the matrix logarithm $\ln(I - A)$ via its Taylor–Mercator series, valid when $\rho(A) < 1$ (spectral radius less than 1) or, more conservatively, $\|A\| < 1$ in any sub-multiplicative norm:

$$\ln(I - A) = -A - \frac{1}{2}A^2 - \frac{1}{3}A^3 - \cdots. \tag{44}$$

Taking the trace of (44) gives

$$\operatorname{tr}\ln(I - A) = -\operatorname{tr}(A) - \frac{1}{2}\operatorname{tr}(A^2) - \frac{1}{3}\operatorname{tr}(A^3) - \cdots. \tag{45}$$

If $A$ is small in norm, the higher-order terms $\operatorname{tr}(A^k)$ for $k \geq 2$ scale like $O(\|A\|^2)$ or smaller. Keeping only the leading term yields the first-order approximation:

$$\ln\det(I - A(z)) \approx -\operatorname{tr}(A(z)), \tag{46}$$

with an error of order $O(\|A(z)\|^2)$. Substituting this into (41) gives

$$I(X_{l+1}^{t+1}; X_l^{t+1} \mid X_{l+1}^t) \approx \tfrac{1}{2}\operatorname{tr}(A(z)), \tag{47}$$

which has the simple interpretation of being proportional to the total fractional reduction in the innovation variance of $u$ due to conditioning on $(v, y)$.

However, computing the corresponding covariance blocks exactly would require backpropagation through all hidden units and scales poorly with the hidden dimensionality of LLMs and VLMs.

## C. Proof of Theorem 4.3

We start from (15) and show that, under a finite-difference, rank-one approximation to the local Jacobians, the trace $\operatorname{tr}(A(z))$ reduces to a squared cosine similarity.

**From Jacobian to finite differences.** Let $\mathbf{z}_l = h_l(x)$ and $\mathbf{z}_{l+1} = h_{l+1}(x)$ denote the pre-update activations of layers $l$ and $l+1$ for some input $x$. For two nearby inputs $x_i$ and $x_{i+1}$, a first-order Taylor expansion gives

$$\mathbf{z}_l(x_{i+1}) - \mathbf{z}_l(x_i) \approx J_l(x_i)\,\delta x_i, \qquad \mathbf{z}_{l+1}(x_{i+1}) - \mathbf{z}_{l+1}(x_i) \approx J_{l+1}(x_i)\,\delta x_i, \tag{48}$$

where $J_l(x_i)$ and $J_{l+1}(x_i)$ are the Jacobians of layers $l$ and $l+1$ with respect to the input, and $\delta x_i = x_{i+1} - x_i$. In practice, we do not form $J_l$ explicitly; instead, we use the observed finite differences

$$\Delta\mathbf{z}_{l,i} = \mathbf{z}_{l,i+1} - \mathbf{z}_{l,i} \approx J_l(x_i)\,\delta x_i, \tag{49}$$

$$\Delta\mathbf{z}_{l+1,i} = \mathbf{z}_{l+1,i+1} - \mathbf{z}_{l+1,i} \approx J_{l+1}(x_i)\,\delta x_i, \tag{50}$$

as surrogates for the Jacobian–input products $J_l\,\delta x_i$ and $J_{l+1}\,\delta x_i$. This viewpoint justifies interpreting the local covariance matrices in TE as being dominated by the outer products of these difference vectors.

**Rank-one surrogates.** For a fixed $i$, we therefore approximate the local covariances by

$$\Sigma_{uu}^{(i)} \approx \Delta\mathbf{z}_{l+1,i}\,\Delta\mathbf{z}_{l+1,i}^\top + \epsilon I, \tag{51}$$

$$\Sigma_{uz}^{(i)} \approx \Delta\mathbf{z}_{l+1,i}\left(\Delta\mathbf{z}_{l,i}^\top \quad \Delta\mathbf{z}_{l+1,i}^\top\right), \tag{52}$$

$$\Sigma_{zz}^{(i)} \approx \begin{pmatrix} \Delta\mathbf{z}_{l,i}\Delta\mathbf{z}_{l,i}^\top & \Delta\mathbf{z}_{l,i}\Delta\mathbf{z}_{l+1,i}^\top \\ \Delta\mathbf{z}_{l+1,i}\Delta\mathbf{z}_{l,i}^\top & \Delta\mathbf{z}_{l+1,i}\Delta\mathbf{z}_{l+1,i}^\top \end{pmatrix} + \epsilon I, \tag{53}$$

where $\epsilon I$ provides a small ridge for numerical stability. As in standard rank-one whitening, $\Sigma_{uu}^{(i)\,-1/2}$ projects mainly onto the normalized direction of $\Delta\mathbf{z}_{l+1,i}$, and a small-coupling approximation renders $\Sigma_{zz}^{(i)\,-1}$ nearly block-diagonal with scalings $(\|\Delta\mathbf{z}_{l,i}\|^2 + \epsilon)^{-1}$ and $(\|\Delta\mathbf{z}_{l+1,i}\|^2 + \epsilon)^{-1}$ on each block.

**Trace evaluation.** Substituting the rank-one forms into

$$A^{(i)} = \Sigma_{uu}^{(i)\,-1/2}\,\Sigma_{uz}^{(i)}\,\Sigma_{zz}^{(i)\,-1}\,\Sigma_{zu}^{(i)}\,\Sigma_{uu}^{(i)\,-1/2} \tag{54}$$

lets us evaluate $\operatorname{tr}(A^{(i)})$ explicitly.

First, note that $\Sigma_{uu}^{(i)} \approx \Delta\mathbf{z}_{l+1,i}\,\Delta\mathbf{z}_{l+1,i}^\top + \epsilon I$ is a rank-one perturbation of $\epsilon I$. Its inverse square root projects primarily onto the normalized direction $\hat{\mathbf{u}} = \Delta\mathbf{z}_{l+1,i}/\|\Delta\mathbf{z}_{l+1,i}\|$, scaled by $(\|\Delta\mathbf{z}_{l+1,i}\|^2 + \epsilon)^{-1/2}$, plus an orthogonal component that does not contribute under the rank-one construction.

Next,

$$\Sigma_{uz}^{(i)} \approx \Delta\mathbf{z}_{l+1,i} \left( \Delta\mathbf{z}_{l,i}^\top \quad \Delta\mathbf{z}_{l+1,i}^\top \right), \tag{55}$$

and $\Sigma_{zz}^{(i)\,-1}$ is approximately block-diagonal with scalings $(\|\Delta\mathbf{z}_{l,i}\|^2 + \epsilon)^{-1}$ and $(\|\Delta\mathbf{z}_{l+1,i}\|^2 + \epsilon)^{-1}$ on the two diagonal blocks, since cross-terms are negligible under the weak-coupling assumption.

Multiplying (55) by $\Sigma_{zz}^{(i)\,-1}$ yields

$$\Sigma_{uz}^{(i)}\Sigma_{zz}^{(i)\,-1} \approx \Delta\mathbf{z}_{l+1,i} \left( \frac{\Delta\mathbf{z}_{l,i}^\top}{\|\Delta\mathbf{z}_{l,i}\|^2 + \epsilon}, \; \frac{\Delta\mathbf{z}_{l+1,i}^\top}{\|\Delta\mathbf{z}_{l+1,i}\|^2 + \epsilon} \right). \tag{56}$$

Right-multiplying (56) by $\Sigma_{zu}^{(i)}$ (the transpose of $\Sigma_{uz}^{(i)}$) produces

$$\Sigma_{uz}^{(i)}\Sigma_{zz}^{(i)\,-1}\Sigma_{zu}^{(i)} \approx \frac{\Delta\mathbf{z}_{l+1,i}\,\Delta\mathbf{z}_{l,i}^\top\,\Delta\mathbf{z}_{l,i}\,\Delta\mathbf{z}_{l+1,i}^\top}{(\|\Delta\mathbf{z}_{l,i}\|^2 + \epsilon)(\|\Delta\mathbf{z}_{l+1,i}\|^2 + \epsilon)}. \tag{57}$$

This is again rank-one, aligned with $\widehat{\mathbf{u}}$, with scalar weight

$$\frac{\left( \Delta\mathbf{z}_{l+1,i}^\top\Delta\mathbf{z}_{l,i} \right)^2}{(\|\Delta\mathbf{z}_{l,i}\|^2 + \epsilon)(\|\Delta\mathbf{z}_{l+1,i}\|^2 + \epsilon)}. \tag{58}$$

Finally, sandwiching (57) with $\Sigma_{uu}^{(i)\,-1/2}$ from (54) divides out the $(\|\Delta\mathbf{z}_{l+1,i}\|^2 + \epsilon)$ term in (58), leaving

$$\mathrm{tr}\!\left(A^{(i)}\right) \approx \frac{\left( \Delta\mathbf{z}_{l+1,i}^\top\Delta\mathbf{z}_{l,i} \right)^2}{\|\Delta\mathbf{z}_{l+1,i}\|^2\,\|\Delta\mathbf{z}_{l,i}\|^2 + \epsilon}. \tag{59}$$

Recognizing the normalized dot product is a cosine similarity, (59) becomes

$$\mathrm{tr}\!\left(A^{(i)}\right) \approx \cos^2\!\left(\Delta\mathbf{z}_{l+1,i},\, \Delta\mathbf{z}_{l,i}\right) + O(\epsilon). \tag{60}$$

Averaging over $i = 1, \ldots, B-1$ and multiplying by $1/2$ (per (15)) yields (18), and averaging over batches further reduces estimator variance. The TE estimation is obtained

$$\mathrm{I}\!\left(X_{l+1}^{t+1};\, X_l^{t+1} \mid X_{l+1}^t\right) \approx \frac{1}{2(B-1)} \sum_{i=1}^{B-1} \cos^2\!\left(\Delta\mathbf{z}_{l+1,i},\, \Delta\mathbf{z}_{l,i}\right), \tag{61}$$

## D. Validity and Error Control for TE Surrogates

A central challenge in applying TE to LLM/VLM hidden states is that the exact conditional mutual information in (3) is intractable at scale. Our method therefore uses a sequence of approximations: (i) a *local linear–Gaussian* model around a probe operating point (Theorem 4.1), (ii) a *first-order* (trace) truncation of the resulting log-determinant TE (Theorem 4.1), and (iii) a *rank-one, finite-difference* surrogate that yields a bounded cosine proxy (Theorem 4.3). This section clarifies what each approximation estimates, provides explicit error bounds for the trace truncation, and describes empirical diagnostics we use to validate that the proxy preserves *ordering* and is robust enough for pruning decisions.

### D.1. What our estimator approximates (and what it does not)

Theorem 4.1 characterizes TE under a local linear response model driven by a small isotropic probe perturbation $\delta x \sim \mathcal{N}(0, \sigma^2 I)$. The resulting quantity is a *local linear–Gaussian TE* that depends on the Jacobian-induced covariances at the operating point. The cosine estimator in Theorem 4.3 is a further simplification that replaces $J\delta x$ with *finite differences* across the probe batch and imposes a rank-one covariance approximation. Accordingly, throughout the paper we use $\widehat{\mathrm{TE}}_\ell(t)$ primarily as a *relative indicator* (for ranking/pruning), rather than claiming it recovers the exact $\mathrm{TE}_\ell(t)$ in nats for the full nonlinear model.

**D.2. Trace truncation error for the linear–Gaussian TE**

Under the linear–Gaussian model (Theorem 4.1), TE admits

$$\text{TE}_{\text{LG}} = -\tfrac{1}{2} \log \det(I - A), \qquad A \succeq 0, \ \rho(A) < 1, \tag{62}$$

where $\rho(\cdot)$ is spectral radius. Using the standard series $-\log \det(I - A) = \sum_{k \geq 1} \tfrac{1}{k} \text{tr}(A^k)$, the trace surrogate corresponds to keeping only the $k = 1$ term:

$$\text{TE}_{\text{trace}} = \tfrac{1}{2} \text{tr}(A). \tag{63}$$

Let $\lambda_1, \ldots, \lambda_m$ denote the eigenvalues of $A$ (all in $[0, 1)$). Then the truncation error has an exact series form

$$\text{TE}_{\text{LG}} - \text{TE}_{\text{trace}} = \tfrac{1}{2} \sum_{i=1}^{m} \sum_{k \geq 2} \frac{\lambda_i^k}{k} \ \geq \ 0, \tag{64}$$

i.e., the trace surrogate is *conservative* (it underestimates TE). Moreover, it admits a non-asymptotic bound:

$$0 \leq \text{TE}_{\text{LG}} - \text{TE}_{\text{trace}} \leq \frac{1}{4} \sum_{i=1}^{m} \frac{\lambda_i^2}{1 - \lambda_i} \ \leq \ \frac{1}{4} \cdot \frac{\|A\|_F^2}{1 - \|A\|_2}, \tag{65}$$

where $\|\cdot\|_2$ and $\|\cdot\|_F$ are the spectral and Frobenius norms. Eq. (65) provides an explicit *a posteriori* criterion: when $\|A\|_2$ is bounded away from 1 and $\|A\|_F$ is small, the trace approximation is accurate. In Appendix G, we empirically verify this conservative behavior and trend preservation by comparing $\text{TE}_{\text{trace}}$ and the exact log-det TE under controlled low-rank sweeps.

**Practical implication.** Our pruning rule uses $\widehat{\text{TE}}$ as a *ranking score*. Even when $\text{TE}_{\text{trace}}$ is biased low, Eq. (64) shows the bias increases smoothly with the eigenvalues of $A$, which preserves ordering in regimes where the spectrum is not near 1. We additionally stabilize the score with EMA smoothing and subtract a null baseline (21), further reducing sensitivity to small absolute errors.

**D.3. Local linearization error and perturbation-scale checks**

The linear–Gaussian model itself is a local approximation. Write the post-update responses as

$$u = u_0 + J_{l+1} \delta x + r_u(\delta x), \qquad v = v_0 + J_l \delta x + r_v(\delta x), \tag{66}$$

where $r_u, r_v$ are Taylor remainders. If the Jacobians are locally Lipschitz in a neighborhood of the probe point (equivalently, the relevant Hessians are bounded), then there exist constants $C_u, C_v$ such that

$$\|r_u(\delta x)\| \leq C_u \|\delta x\|^2, \qquad \|r_v(\delta x)\| \leq C_v \|\delta x\|^2. \tag{67}$$

Thus, for sufficiently small $\sigma$, the remainder energy is $O(\sigma^4)$ whereas the linear term energy is $O(\sigma^2)$, implying the linear–Gaussian TE is stable to higher-order nonlinearities in the *small-perturbation* regime. In practice, we treat $\sigma$ as a hyperparameter and verify stability by a *scale sweep*: we require the layerwise TE ranking to be consistent over a range of small $\sigma$ (before numerical noise dominates at extremely small $\sigma$).

**Takeaway.** While our cosine estimator is not claimed to recover exact TE in the full nonlinear model, we provide (i) explicit error control for the log-det $\to$ trace truncation Eq. (65), (ii) a local linearization regime with perturbation-scale checks Eq. (67).

# E. Summary of TE Approximation Algorithm

We summarize the procedure for computing instantaneous, layerwise transfer entropy using the cosine–similarity approximation in Algorithm 1. The method evaluates a fixed probe batch before and after each training update, forms finite-difference activation vectors for adjacent layers, and estimates TE via squared cosine similarity under a rank-one covariance approximation.

---

**Algorithm 1** Instantaneous Layerwise TE via Cosine–Similarity (single combined procedure)

---

**Require:** Model with layers $1\ldots L$; training steps $t = 1\ldots T$; fixed probe batch $\mathcal{B}$; ridge $\epsilon > 0$; window size $W$; (optional) pooling operator $\text{Pool}(\cdot)$.

**Ensure:** Curves $\widehat{\text{TE}}_\ell(t)$ and windowed averages $\overline{\text{TE}}_\ell(t)$ for all $\ell = 1, \ldots, L-1$ and $t = 1, \ldots, T-1$.

1: **for** $t = 1$ to $T - 1$ **do**
2:      Run forward on $\mathcal{B}$ with $\theta_t$ and cache $h_t^\ell$ for all $\ell$.
3:      Apply one optimizer step to obtain $\theta_{t+1}$.
4:      Re-run forward on $\mathcal{B}$ with $\theta_{t+1}$ and cache $h_{t+1}^\ell$.
5:      **for** $\ell = 1$ to $L - 1$ **do** {last layer has no upper neighbor}
         **(A) Select layer-pair tensors**
6:          $V \leftarrow h_{t+1}^\ell$ {$X_\ell^{t+1}$: post-update, lower layer}
7:          $Y \leftarrow h_t^{\ell+1}$ {$X_{\ell+1}^t$: pre-update, upper layer}
8:          $U \leftarrow h_{t+1}^{\ell+1}$ {$X_{\ell+1}^{t+1}$: post-update, upper layer}
         **(B) Optional pooling (token/sequence level)**
9:          **if** pooling enabled **then**
10:            $V \leftarrow \text{Pool}(V); \; Y \leftarrow \text{Pool}(Y); \; U \leftarrow \text{Pool}(U)$
11:          **end if**
         **(C) Finite differences across mini-batch order**
12:          Let $B$ be the number of (pooled) items in the batch.
13:          **for** $i = 1$ to $B - 1$ **do**
14:            $\Delta\mathbf{z}_{\ell,i} \leftarrow V_{i+1} - V_i$
15:            $\Delta\mathbf{z}_{\ell+1,i} \leftarrow U_{i+1} - U_i$
16:          **end for**
         **(D) Cosine-squared aggregation with ridge**
17:          $s \leftarrow 0$
18:          **for** $i = 1$ to $B - 1$ **do**
19:            $n_i \leftarrow \|\Delta\mathbf{z}_{\ell+1,i}\|^2 \cdot \|\Delta\mathbf{z}_{\ell,i}\|^2 + \epsilon$
20:            $c_i \leftarrow \dfrac{\left(\Delta\mathbf{z}_{\ell+1,i}^\top \Delta\mathbf{z}_{\ell,i}\right)^2}{n_i}$
21:            $s \leftarrow s + c_i$
22:          **end for**
         **(E) Instantaneous TE estimate (per (15))**
23:          $\widehat{\text{TE}}_\ell(t) \leftarrow \dfrac{1}{2(B-1)} s$
24:      **end for**
25: **end for**
26: **(F) Temporal smoothing (optional)**: For all $\ell$, set $\overline{\text{TE}}_\ell(t) \leftarrow \frac{1}{W} \sum_{i=0}^{W-1} \widehat{\text{TE}}_\ell(t-i)$ for $t = W, \ldots, T-1$.
27: **return** $\{\widehat{\text{TE}}_\ell(t)\}$ and (optionally) $\{\overline{\text{TE}}_\ell(t)\}$ for all layers $\ell$ and steps $t$.

---

## F. Computational Complexity of TE and TE Approximation

**Notation and cost model.** Let $N$ denote the number of items used by the estimator at a given step (e.g., $N = $ sequences in the probe batch), and let $d_l, d_{l+1}$ be the hidden sizes at layers $l$ and $l+1$. We write $d := d_l + d_{l+1}$ and $d_u = d_{l+1}$. Let $L$ be the number of layers whose TE is evaluated per step. One evaluation of the model's forward pass on the probe batch is denoted by $C_{\text{fwd}}$ (wall-clock FLOPs/latency). Unless stated, complexities below are *per layer, per step*, not counting the two forward passes already needed to cache $\{y, v, u\}$ (Section 4).

### F.1. Exact TE and density-based estimators (impractical at scale)

The definition $\text{TE}_l(t) = I(u; v \mid y)$ requires densities for $(u, v, y) \in \mathbb{R}^{d_u + d_l + d_u}$, i.e., a $D$-dimensional joint with $D = d_l + 2d_{l+1}$. Nonparametric conditional MI estimators (e.g., $k$NN/Kraskov-type) need nearest-neighbor queries in $D$ dimensions:

- $k$**NN CMI**: Naïve pairwise distances: $O(N^2 D)$ time and $O(ND)$ memory; space-partition trees degrade in high $D$, often reverting to near-quadratic time. Additionally, per-sample marginal neighbor counts (required by CMI) induce extra $O(N)$ book-keeping, keeping overall complexity $O(N^2 D)$ in practice.

- **Kernel density CMI**: Bandwidth selection and $N^2$ kernel accumulations yield $O(N^2 D)$ time (or $O(ND)$ with strongly compressed/random-feature approximations), with high variance in large $D$.

In modern VLMs where $d_l, d_{l+1} \sim 10^3$ and $N$ is in the $10^3$–$10^4$ range (tokens), these methods are computationally and statistically prohibitive.

### F.2. Linear–Gaussian (Jacobian) approximation

Under the local linear–Gaussian model (Theorem 4.1), TE reduces to matrix covariances and a small matrix inverse on the concatenated variable $z = (v, y)$. Two cost components dominate:

**(a) Forming covariances.** Given the $N \times d$ design matrices of $(v, y)$ and the $N \times d_u$ for $u$, empirical covariances can be computed via GEMMs in

$$O(Nd^2) \quad \text{for } \Sigma_{zz} \in \mathbb{R}^{d \times d}, \qquad O(Ndd_u) \quad \text{for } \Sigma_{uz} \in \mathbb{R}^{d_u \times d}.$$

Memory is $O(Nd)$ if streamed (Welford/online covariance) or $O(Nd)$–$O(Nd_u)$ if materialized.

**(b) Small dense linear algebra.** Inverting $\Sigma_{zz}$ and forming $A(z)$ (refer to 13) costs

$$O(d^3) \text{ (Cholesky + solves)} + O(d^2 d_u) \text{ (products).}$$

For $d$ in the low thousands this is tractable on GPU, and independent of $N$ once covariances are formed.

### F.3. Variational MI bounds (MINE/InfoNCE-style)

Variational estimators optimize a critic $T_\phi$ to lower/upper-bound $\mathrm{I}(u; v \mid y)$: per update the complexity is $O(N P)$ for a critic with $P$ parameters, plus the cost of negative sampling (often $O(N)$ or $O(N \log N)$; worst-case $O(N^2)$ if all-pairs are used). Over $E$ training steps, time is $O(E N P)$ and memory tracks the critic activations. These methods are significantly heavier than the linear–Gaussian route and introduce extra variance/bias from optimization.

### F.4. Cosine–similarity TE approximation (ours)

The estimator in Theorem 4.3 only needs finite differences and cosine similarities:

$$\text{Time:} \quad O(N(d_l + d_{l+1})), \qquad \text{Space:} \quad O(d_l + d_{l+1}) \text{ (streaming).}$$

It adds negligible overhead beyond the two forward passes used to cache $(y, v, u)$. Because it avoids neighbor searches, Jacobians, and critic training, it scales linearly in both $N$ and dimensionality.

### F.5. Wall-clock and practical considerations

Let $L_{\text{eval}} \le L$ be the number of layers at which TE is computed per step (possibly with stride). Excluding the two standard forward passes to cache $(y, v, u)$, the additional per-step costs are

$$
\begin{aligned}
k\text{NN CMI:} &\quad O\big(L_{\text{eval}} N^2 (d_l + 2d_{l+1})\big), \\
\text{Linear–Gaussian:} &\quad O\big(L_{\text{eval}} [N(d_l + d_{l+1})^2 + (d_l + d_{l+1})^3]\big), \\
\text{Variational ($E$ critic steps):} &\quad O\big(L_{\text{eval}} E N P\big), \\
\text{Cosine TE (ours):} &\quad O\big(L_{\text{eval}} N(d_l + d_{l+1})\big).
\end{aligned}
$$

In multi-head attention where $d_l = d_{l+1} = d_{\text{model}}$, the cosine estimator scales as $O(L_{\text{eval}} N d_{\text{model}})$ and is typically dominated by the baseline forwards. In Table 3, we provide the comparison of asymptotic costs (per layer, per step), excluding the two forward passes used to cache $(y, v, u)$ for different estimators.

*Table 3.* Asymptotic costs (per layer, per step), excluding the two forward passes used to cache $(y, v, u)$.

| Estimator | Time per layer | Space | Comments |
|---|---|---|---|
| $k$NN CMI | $O(N^2(d_l+2d_{l+1}))$ | $O(N(d_l+2d_{l+1}))$ | Degrades in high $D$ |
| Linear–Gaussian (cov.) | $O\big(N(d_l+d_{l+1})^2 + (d_l+d_{l+1})^3\big)$ | $O\big((d_l+d_{l+1})^2\big)$ | Stable; closed form |
| Variational (MINE/InfoNCE) | $O(E\,N\,P)$ | $O(P)$ | Training overhead |
| Cosine TE (ours) | $O(N(d_l+d_{l+1}))$ | $O(d_l+d_{l+1})$ | Best cost/benefit |

**Memory, streaming, and smoothing.** All estimators benefit from streaming reductions that avoid materializing $N \times d$ activations; dot-products, norms, and covariance accumulators can be maintained online in $O(d)$ to $O(d^2)$ memory. To stabilize estimates across steps, we apply Exponential Moving Average (EMA) smoothing to each layer's TE time series,

$$\widehat{\mathrm{TE}}_l^{(\tau)} \leftarrow \alpha\, \widehat{\mathrm{TE}}_l^{(\tau-1)} + (1 - \alpha)\, \mathrm{TE}_l^{(\tau)}, \tag{68}$$

which adds $O(L_{\mathrm{eval}})$ time and $O(L_{\mathrm{eval}})$ memory per step.

**Practical guidance.** For online diagnostics and pruning rules under tight budgets, the cosine–similarity estimator offers linear time and negligible memory overhead. When post-hoc analysis is feasible, the linear–Gaussian approximation provides calibrated, closed-form values at $O(Nd^2+d^3)$ cost. Nearest-neighbor or kernel CMI degrades sharply in high dimensions and is advisable only for very low-dimensional probes, while variational bounds, though flexible, incur the highest computational and optimization overhead and are best reserved for small-scale validation.

## G. TE Approximation versus Exact TE: Performance Comparison

Following Section 4, we linearize the two adjacent layers around a fixed probe batch and inject a small Gaussian perturbation, yielding $u \approx J_{l+1}\delta x + \varepsilon_u$, $v \approx J_l\delta x + \varepsilon_v$ with $\delta x \sim \mathcal{N}(0, \sigma^2 I)$ and isotropic noises $\varepsilon_u, \varepsilon_v$. Since $y = X_{l+1}^t$ is fixed at the operating point, we have $I(u; v \mid y) = I(u; v)$. We study a *latent rank sweep* by varying the number of driven directions $r \in \{1, \ldots, 8\}$ at a fixed, high alignment between the left singular subspaces of $J_l$ and $J_{l+1}$. We define the latent rank $r$ as the dimension of the driven subspace $\delta x \in \mathbb{R}^r$ that feeds both layers, equivalently $\mathrm{rank}(\Sigma_{uv})$, i.e., the number of nonzero canonical correlations between $u$ and $v$ that contribute to TE.

Let

$$A = \Sigma_{uu}^{-1/2}\, \Sigma_{uv}\, \Sigma_{vv}^{-1}\, \Sigma_{vu}\, \Sigma_{uu}^{-1/2}, \tag{69}$$

with covariance blocks $\Sigma_{uu}, \Sigma_{vv}, \Sigma_{uv}$ implied by $J_{l+1}, J_l$ under the local model. We compare:

- **Exact TE**:

$$\mathrm{TE}_{\mathrm{exact}} = -\tfrac{1}{2} \ln \det\big(I - A\big). \tag{70}$$

- **Trace approximation (Theorem 4.1).** Using the first term in the series for $-\ln\det(I - A)$,

$$\mathrm{TE}_{\mathrm{trace}} = \tfrac{1}{2}\,\mathrm{tr}(A), \tag{71}$$

  which omits higher-order powers of $A$.

- **Cosine–similarity approximation (Theorem 4.3)** Over the ordered probe batch, form finite differences $\Delta u_i = u_{i+1} - u_i$, $\Delta v_i = v_{i+1} - v_i$ and define

$$\widehat{\mathrm{TE}}_{\mathrm{cos}} = \overline{\cos^2\big(\Delta u_i, \Delta v_i\big)} \in [0, 1], \tag{72}$$

  a bounded geometric proxy that we *do not* map into nats for the main comparison.

To compare on a common $[0, 1]$ scale, all TE-in-nats curves (Exact, Trace) are plotted using

$$\mathrm{norm}(\mathrm{TE}) = \frac{\log\big(1 + \mathrm{TE}\big)}{\log\big(1 + \mathrm{TE}_{\mathrm{max}}\big)}, \tag{73}$$

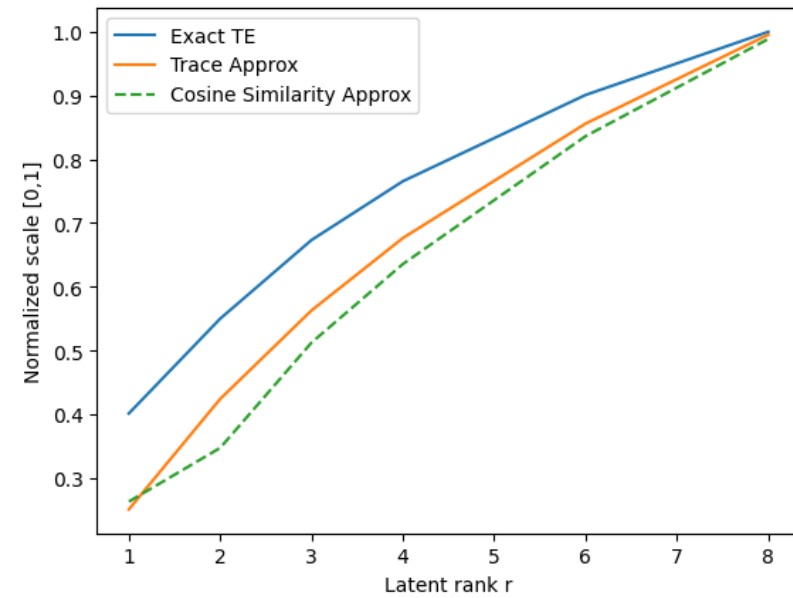

*Figure 3.* Normalized comparison between Exact TE, Trace (Theorem 4.1), and Cosine Similarity-based (Theorem 4.3) approximations as $r$ increases.

where $\mathrm{TE}_{\max}$ is the maximum $\mathrm{TE}_{\mathrm{exact}}$ within the sweep. The raw cosine proxy is already bounded and is shown directly as $\widehat{\mathrm{TE}}_{\cos} \in [0, 1]$.

In Fig. 3, we summarize the normalized results of exact TE, and approximated TE from Theorems 4.1 and 4.3. Observe Fig. 3,

1. Monotone growth with rank. The exact TE increases steadily with $r$, reflecting the additive contribution of multiple driven directions shared by $J_l$ and $J_{l+1}$.

2. Trace is conservative but trend-accurate. The first-order trace surrogate (71) systematically *underestimates* the exact TE across ranks while closely tracking its shape; the gap narrows as $r$ increases, indicating moderate higher-order corrections.

3. Cosine-similarity preserves ordering. The raw cosine proxy rises smoothly with $r$ and preserves the rank ordering, with a larger downward bias at small $r$ that diminishes toward higher ranks.

Under the Jacobian Linear–Gaussian model, latent rank is a primary driver of layerwise transfer entropy. The cosine approximation computed from finite differences on cached activations provides a low-cost and stable *relative* indicator of TE changes across ranks, making it suitable for monitoring layerwise information flow during fine-tuning. For *absolute* quantification and theory alignment, the determinant form (70) is preferred; the first-order trace surrogate (71) is a useful, and cosine-similarity approximation works well.

## H. Additional Experimental Results for LLMs

### H.1. Comparison with Knowledge Entropy

While both our TE and the Knowledge Entropy (KE) of Kim et al. (Kim et al., 2025) aim to quantify the information dynamics within deep language models, they capture fundamentally different aspects of model behavior. The knowledge entropy measures the uncertainty or variability in how a language model utilizes its parametric knowledge, specifically focusing on how broadly the model integrates memory vectors (such as the values in feed-forward layers) via the distribution of memory coefficients. This concept reflects the model's reliance on a diverse versus narrow set of knowledge sources during pretraining, and its decline is shown to impede new knowledge acquisition and retention. In contrast, our TE

formalism measures the conditional uncertainty in the representations produced by each layer during fine-tuning, explicitly modeling how the output of a layer at a given epoch depends on previous representations across both space (layers) and time (epochs). Thus, while knowledge entropy pertains to the diversity of internal memory utilization, TE characterizes the information dynamics of model representations as they evolve over training and through the network's depth, offering a broader space-time view of uncertainty propagation in LLMs.

In parallel with the TE analysis, we computed the knowledge entropy at each layer of RoBERTa$_{base}$ throughout full fine-tuning. As shown in Fig. 4, the knowledge entropy consistently exhibits a monotonic decrease from the input to the output layers, with only minor fluctuations across epochs. The initial layers maintain high knowledge entropy values (e.g., above 3.2), which steadily decline as information propagates deeper into the model, reaching their lowest levels (as low as 2.7) in the final output layers. This trend is remarkably robust, with the overall shape and absolute values of the knowledge entropy profile remaining nearly invariant across training epochs. The observed monotonic decay reflects the progressive refinement and compression of the input representation, with uncertainty and information content being gradually distilled as the model approaches its final prediction.

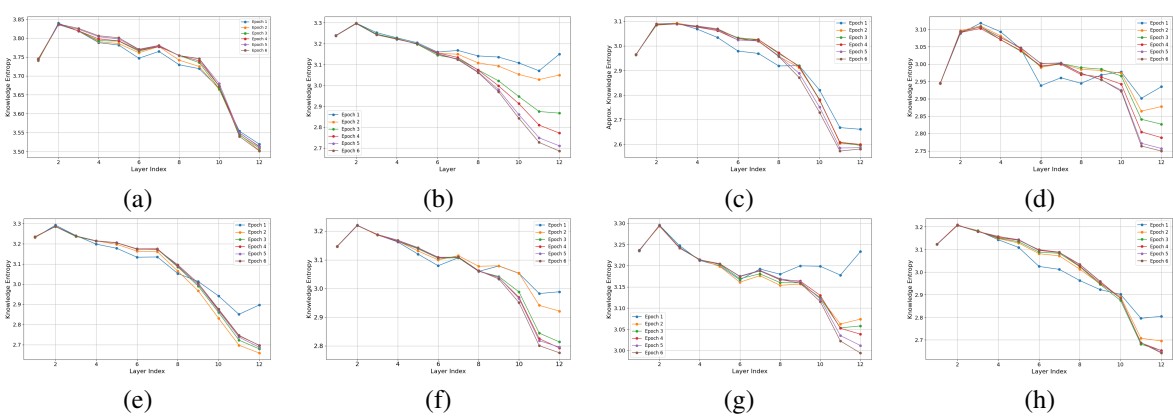

*Figure 4.* The knowledge entropy versus layer index in RoBERTa$_{base}$ with full fine tuning for 6 epochs using eight datasets on the GLUE benchmark, (a) MNLI, (b) MRPC, (c) SST-2, (d) CoLA, (e) QNLI, (f) QQP, (g) RTE, (h) STS-B.

Contrasting this behavior with the layerwise TE profile reveals key differences in how these two measures characterize model dynamics. While knowledge entropy captures the static uncertainty of representations and reflects the gradual consolidation of knowledge along the depth of the network, the TE provides a dynamic perspective by quantifying the flow of information between adjacent layers. Notably, the TE displays a pronounced non-monotonic "U-shaped" curve, peaking in the middle layers and indicating zones of elevated representational flexibility and information exchange. In contrast, knowledge entropy decreases smoothly and does not capture these dynamic transitions. Thus, TE uncovers a functional specialization within the network, highlighting regions of heightened adaptation and integration that are obscured by the more static, cumulative view offered by knowledge entropy. This comparison underscores the value of TE as a complementary metric for understanding the layerwise information flow of transformer models during fine-tuning.

## H.2. TE in LoRA-Based Fine Tuning of LLMs

In Fig. 5, we plot the TE versus the layer index in RoBERTa$_{base}$ LoRA fine tuning for eight datasets on the GLUE benchmark. Across all six epochs of LoRA-based fine-tuning, we observe that the TE for each transformer layer remains highly stable. Specifically, the TE curves for each epoch are nearly indistinguishable, and the numerical values for each layer fluctuate only minimally from one epoch to the next. This trend holds for both lower and higher layers, indicating that the relative ordering and magnitude of TE across layers are well-preserved throughout the LoRA training process. These results suggest that LoRA adaptation, unlike standard full-parameter fine-tuning, does not substantially alter the internal information dynamics of the network as measured by TE, even as downstream task accuracy improves.

The small standard deviation of layerwise TE observed during LoRA fine-tuning demonstrates that the information-theoretic structure of the model is remarkably robust under LoRA updates. This stability has important practical implications: TE can serve as a reliable and repeatable metric for both model analysis and pruning decisions, even in the presence of continued LoRA-based adaptation. Furthermore, the insensitivity of TE to LoRA's low-rank parameter updates indicates that TE-based

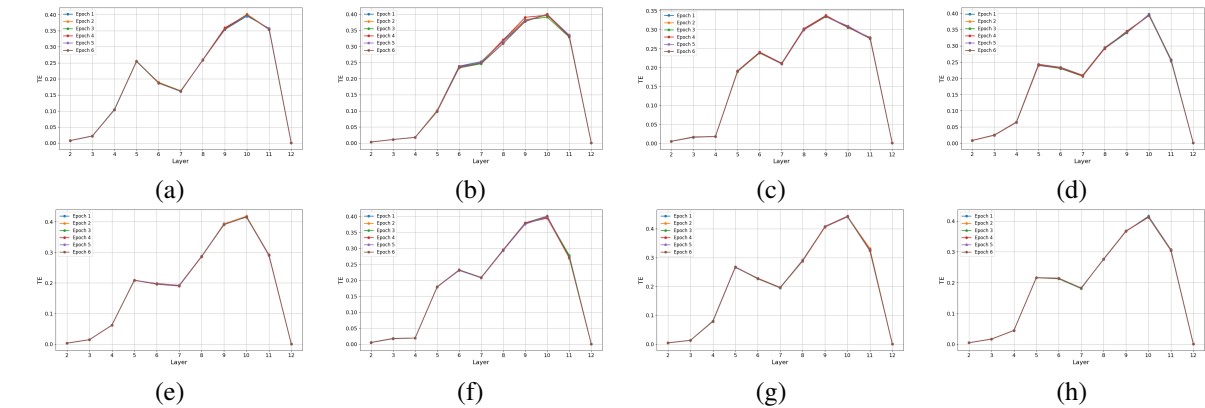

*Figure 5.* The TE versus layer index in RoBERTa$_{base}$ with LoRA ($r = 2$) tuning for 6 epochs using eight datasets on the GLUE benchmark, (a) MNLI, (b) MRPC, (c) SST-2, (d) CoLA, (e) QNLI, (f) QQP, (g) RTE, (h) STS-B.

diagnostics and compression strategies are transferable across epochs, reducing the need for repeated metric estimation at every checkpoint. Overall, these findings reinforce the utility of TE as a principled, stable foundation for layer selection and model optimization in transformer-based architectures.

### H.3. Impact of Learning Rate on Information Flow

The experiments in Section 5.1.1 use a learning rate of $\alpha = 2 \times 10^{-5}$. To isolate the effect of the learning rate on information dynamics during fine-tuning, we repeat the same setup with a substantially larger value, $\alpha = 0.02$, which is unusually high for Adam-based optimization. Unless noted otherwise, all other hyperparameters and data processing are identical to Section 4.1: RoBERTa$_{base}$, full fine-tuning for 6 epochs on the eight GLUE tasks.

Figure 6 shows the evolution of the *layerwise TE* under $\alpha = 0.02$. Across tasks, the first epoch exhibits uniformly high TEs—consistent with large representational updates as the model rapidly adapts. Beginning in epoch 2, however, the scores collapse across nearly all layers and remain low thereafter. Validation accuracy plateaus at the same time, indicating that optimization has converged very quickly—often into a sharp local minimum—with little subsequent representation change. In other words, the model ceases to acquire new information, as reflected by both the suppressed TEs and the stagnant accuracy.

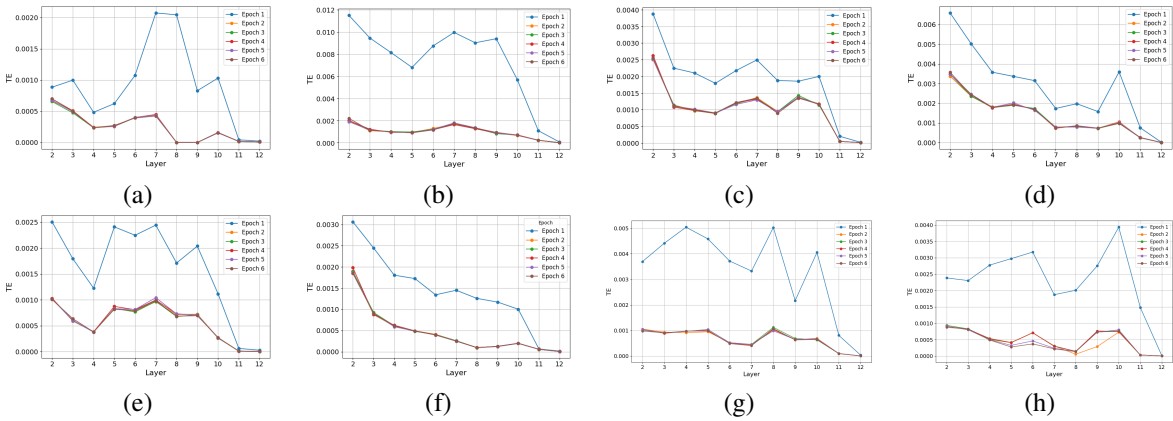

*Figure 6.* With learning rate $\alpha = 0.02$, *TE* versus layer index in RoBERTa$_{base}$ during 6-epoch full fine-tuning on GLUE: (a) MNLI, (b) MRPC, (c) SST-2, (d) CoLA, (e) QNLI, (f) QQP, (g) RTE, (h) STS-B.

By contrast, with the smaller learning rate $\alpha = 2 \times 10^{-5}$ (Section 4.1), we observe a qualitatively different pattern: TEs in intermediate layers remain elevated for multiple epochs, indicating continued updates to internal representations and sustained information flow. This prolonged, non-trivial activity is accompanied by gradual accuracy improvements,

consistent with steadier exploration of the solution space and better generalization.

Table 4 quantifies the effect on downstream performance. The standard learning rate yields consistently strong results (accuracy for MNLI/QNLI/RTE/SST-2/QQP/MRPC, Matthews correlation for CoLA, and Pearson correlation for STS-B), ranging from 0.75 (RTE) to 0.92 (SST-2). In contrast, the high learning rate causes large drops across all tasks—for example, MNLI $0.78 \rightarrow 0.34$, SST-2 $0.92 \rightarrow 0.49$, and QNLI $0.87 \rightarrow 0.51$—and fails to converge on STS-B (NaN).

*Takeaway.* The learning rate strongly governs information dynamics: an excessively large $\alpha$ drives premature convergence and representational collapse (low TEs), stalling learning and harming generalization; a moderate $\alpha$ sustains information flow in intermediate layers and supports gradual, robust improvement. We therefore adopt $\alpha = 2 \times 10^{-5}$ for the remainder of our experiments.

*Table 4.* GLUE performance for RoBERTa$_{\text{base}}$ under full fine-tuning with two learning rates. Metrics are accuracy except CoLA (Matthews correlation) and STS-B (Pearson correlation).

| Learning Rate | MNLI | MRPC | SST-2 | CoLA | QNLI | QQP | RTE | STS-B |
|---|---|---|---|---|---|---|---|---|
| $\alpha = 2 \times 10^{-5}$ | 0.7840 | 0.8480 | 0.9220 | 0.8274 | 0.8731 | 0.8386 | 0.7509 | 0.9075 |
| $\alpha = 0.02$ | 0.3400 | 0.6838 | 0.4908 | 0.6913 | 0.5054 | 0.6318 | 0.5271 | NaN |

### H.4. Information Flow in T5-Base

In addition to RoBERTa$_{\text{base}}$, we also study the Text-to-Text Transfer Transformer (T5) architecture, as presented by Raffel *et al.* (Raffel et al., 2020a), which unifies a broad spectrum of NLP tasks by expressing each as a text-to-text problem. T5 is available in multiple configurations, including T5-Small with roughly 60M parameters and T5-Base with about 220M parameters. For our investigation into TE, we focus on T5-Base.

T5-Base is a transformer-based encoder–decoder architecture comprising 12 encoder layers and 12 decoder layers. Each encoder layer contains a multi-head self-attention sublayer and a position-wise feed-forward network, with residual connections and layer normalization. Decoder layers mirror this structure but include an additional multi-head cross-attention sublayer that attends to the encoder's final hidden states, enabling conditioning on the input representations. Concretely, each decoder layer has three attention modules: masked self-attention (to prevent attending to future tokens), cross-attention (encoder–decoder interaction), and a feed-forward sublayer. In T5-Base, all attention modules employ 12 heads with model dimension 768, enabling complex sequence transduction and flexible conditioning.

We evaluate using four established benchmarks—e-SNLI (Camburu et al., 2018), ANLI (Nie et al., 2019), CommonsenseQA (CQA) (Talmor et al., 2019), and SVAMP (Patel et al., 2021)—chosen to span language understanding and reasoning scenarios. All samples are preprocessed with the T5 tokenizer for consistent training/evaluation. Following (Hsieh et al., 2023), we refer to e-SNLI, CQA, ANLI, and SVAMP as CoT (Chain-of-Thought) benchmarks.

We fine-tune T5-Base separately on each dataset. Figure 7 shows *TE* profiles across decoder layers over six epochs. The curves exhibit a characteristic ∩ shape, peaking at intermediate decoder layers across epochs. This indicates that the intermediate decoder layers are the locus of active information transformation and integration to facilitate output generation and loss minimization. In contrast, the final decoder layer's TE consistently decreases, reflecting the need for stable, deterministic outputs as decoding concludes. These observations highlight a functional specialization: intermediate layers contribute novel, task-relevant transformation, whereas the output layer consolidates representations into predictions.

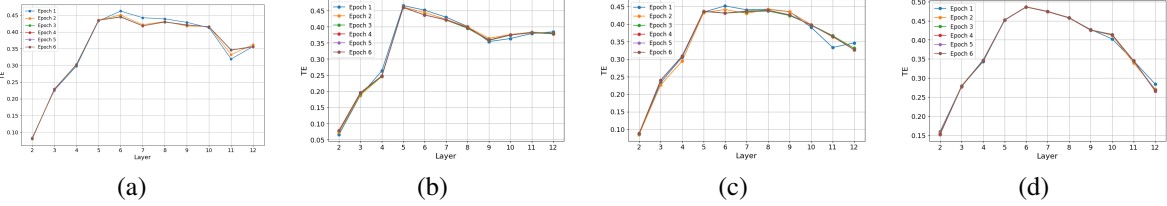

|     (a)     |     (b)     |     (c)     |     (d)     |

*Figure 7.* **Decoder (T5-Base).** TE vs. layer index during 6-epoch full fine-tuning on four datasets: (a) e-SNLI, (b) CQA, (c) ANLI, (d) SVAMP.

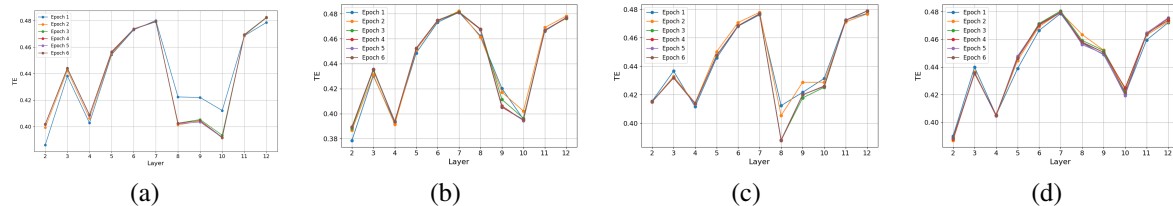

(a)         (b)         (c)         (d)

*Figure 8.* **Encoder (T5-Base).** TE vs. layer index during 6-epoch full fine-tuning on four datasets: (a) e-SNLI, (b) CQA, (c) ANLI, (d) SVAMP.

Figure 8 shows TE as a function of encoder layer index. Unlike the simpler ∩-shaped or monotonic trends often seen in encoder-only models like RoBERTa-Base, T5-Base's encoder exhibits an oscillatory pattern (up → down → up → down → up). We attribute this non-monotonic behavior to encoder–decoder interaction: the presence of a decoder enables iterative information exchange and refinement across stages, repeatedly transforming and integrating representations in the encoder. This interplay fundamentally shapes layer-wise information dynamics in sequence-to-sequence models like T5-Base.

Figure 9 reports *knowledge entropy* (KE) across decoder layers after six epochs on each dataset. The KE profiles typically decrease initially, rise through intermediate layers, and then plateau (down → up → flat).

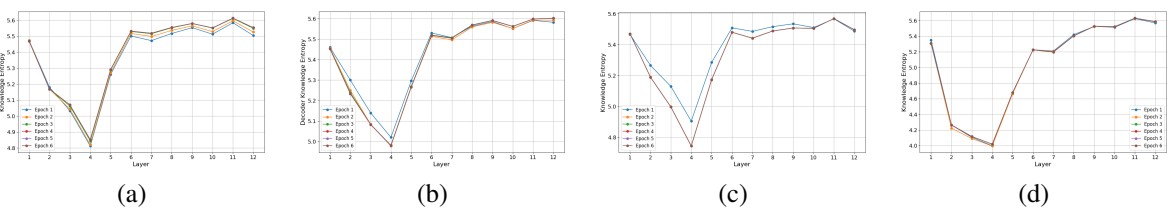

(a)         (b)         (c)         (d)

*Figure 9.* **Decoder (T5-Base).** Knowledge entropy vs. layer index during 6-epoch full fine-tuning on four datasets: (a) e-SNLI, (b) CQA, (c) ANLI, (d) SVAMP.

Figure 10 shows KE across encoder layers. Notably, ANLI exhibits a distinct KE pattern relative to the other datasets, underscoring that KE depends on both architecture and data characteristics.

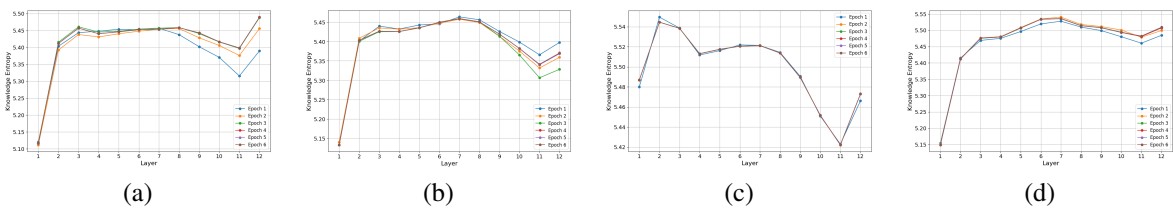

(a)         (b)         (c)         (d)

*Figure 10.* **Encoder (T5-Base).** Knowledge entropy vs. layer index during 6-epoch full fine-tuning on four datasets: (a) e-SNLI, (b) CQA, (c) ANLI, (d) SVAMP.

We next prune four of the twelve decoder layers in T5-Base using four strategies: TE–based pruning, KE-based pruning (KE-Pruning), random LayerDrop, SlimLLM (Guo et al., 2025), and a fine-tuning baseline without pruning. Table 5 compares accuracy on e-SNLI, CQA, ANLI, and SVAMP. Fine-tuning attains the highest accuracy on e-SNLI and CQA, while RS-Pruning performs best on ANLI and SVAMP, with a large margin on SVAMP (0.3400 vs. 0.2600 for fine-tuning). KE-Pruning yields intermediate results, and LayerDrop is lowest across datasets. These findings demonstrate the effectiveness of a principled, TE–driven pruning criterion over random dropout and show that targeted pruning can rival or surpass standard fine-tuning on challenging tasks.

*Table 5.* T5-Base accuracy on four benchmarks.

| Approach | e-SNLI | CQA | ANLI | SVAMP |
|---|---|---|---|---|
| Fine-Tuning | **0.5375** | **0.5807** | **0.3890** | 0.2600 |
| Our TE-Pruning | 0.5350 | 0.5717 | 0.3690 | **0.3400** |
| KE-Pruning | 0.5310 | 0.5405 | 0.3180 | 0.3200 |
| Layer Dropout | 0.5250 | 0.4742 | 0.2900 | 0.3050 |
| SlimLLM | 0.5320 | 0.5667 | 0.3690 | 0.2700 |

## H.5. Model Architecture vs Dataset: Impact on Information Flow

### H.5.1. DATASET IMPACT ON INFORMATION FLOW

To systematically investigate the interplay between model architecture and data domain in shaping information dynamics, we analyze TE using two widely adopted pretrained language models: RoBERTa$_{base}$ and T5-Base. For each model, we fine-tune on a comprehensive suite of datasets representative of standard evaluation benchmarks. Specifically, RoBERTa$_{base}$ is evaluated on the GLUE benchmark, while T5-Base is tested on a diverse set of chain-of-thought (CoT) reasoning datasets. A natural question follows: Are the observed TE profiles an intrinsic property of the model architecture, or do they depend critically on the choice of dataset? In other words, if we exchange the datasets—fine-tuning RoBERTa$_{base}$ on CoT tasks and T5-Base on GLUE—will the resulting TE trajectories remain similar, or will they exhibit fundamentally different patterns?

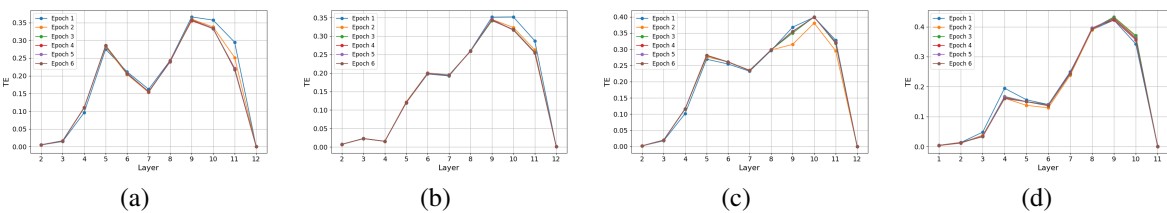

|   (a)   |   (b)   |   (c)   |   (d)   |

*Figure 11.* **RoBERTa$_{base}$ (encoder).** TE vs. layer index during 6-epoch full fine-tuning on four CoT datasets: (a) e-SNLI, (b) CQA, (c) ANLI, (d) SVAMP.

To answer this, we perform experiments for both models under swapped dataset conditions. For each model–dataset pair, we compute the layer-wise TE across all epochs of fine-tuning, thereby capturing the evolution of information dynamics through the model's hierarchy. The results for RoBERTa$_{base}$ on four CoT datasets are shown in Figure 11, while Figures 12 and 13 present the TE profiles for the T5-Base *encoder* and *decoder*, respectively, across the eight GLUE datasets. By comparing these figures, we disentangle the relative contributions of model structure and dataset complexity to the formation of layer-wise information dynamics. This analysis provides insights into the generality and dataset-sensitivity of the TE as a diagnostic for neural information processing in pretrained language models.

In Figure 14, we show the mean and standard deviation of the TE across the 12 encoder layers of RoBERTa$_{base}$, aggregated over 12 datasets and 6 epochs of fine-tuning. The mean profile is strongly non-monotonic: it starts low in the initial layers, rises to a broad peak spanning mid-to-upper layers (layers 4–9), then drops sharply at the output layer. Notably, the highest TEs occur at layers 7–9 (mean $\approx$ 0.24–0.29), indicating these layers sustain the most dynamic representational changes during adaptation. The standard deviation mirrors this pattern, with peak variability in the same layers, suggesting substantial cross-dataset and temporal diversity in information flow at these depths. In contrast, the input and output layers consistently exhibit low scores and minimal variability, reflecting more stable processing roles. These findings highlight the mid-to-upper layers of RoBERTa$_{base}$ as the most dynamic and dataset-sensitive during fine-tuning, underscoring their critical role in task adaptation, while the extremal layers serve as stable feature encoders/decoders.

In Figure 15a, we present the mean and standard deviation of the TE in the T5-Base encoder over 12 layers, aggregated from six epochs across 12 datasets. The profile is notably high and stable: the mean begins near 0.78 in the input layer, quickly rises above 0.94 in the middle layers, and peaks near 0.96 at layers 5–6. This elevated plateau reflects sustained representational activity and information flow, indicating strong capacity for flexible representation and complex feature extraction. The standard deviation remains very low (below 0.03) through the lower/middle layers, demonstrating consistency across datasets, then increases in the upper layers (especially layers 7–9, std $\approx$ 0.06–0.07), suggesting increased sensitivity

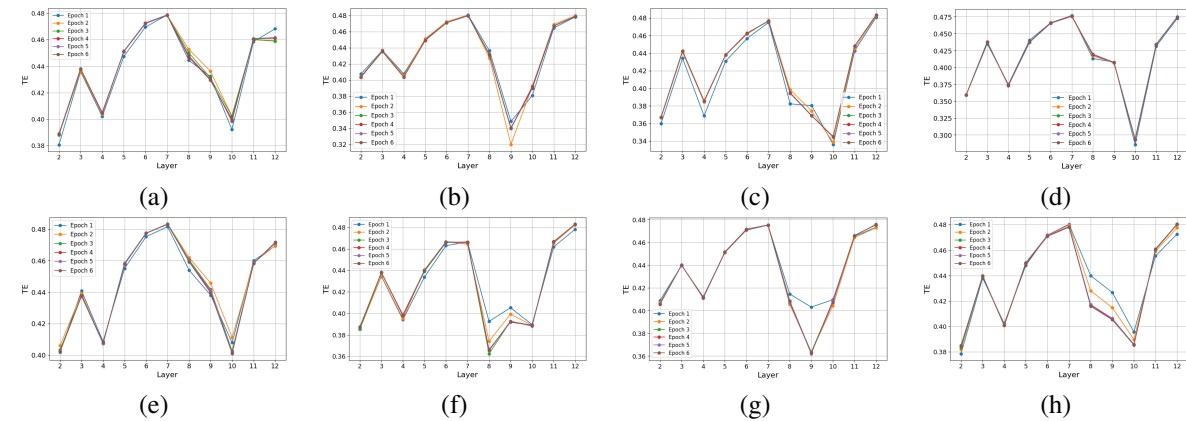

*Figure 12.* **T5-Base (encoder).** TE vs. layer index during 6-epoch full fine-tuning on the GLUE benchmark: (a) MNLI, (b) MRPC, (c) SST-2, (d) CoLA, (e) QNLI, (f) QQP, (g) RTE, (h) STS-B.

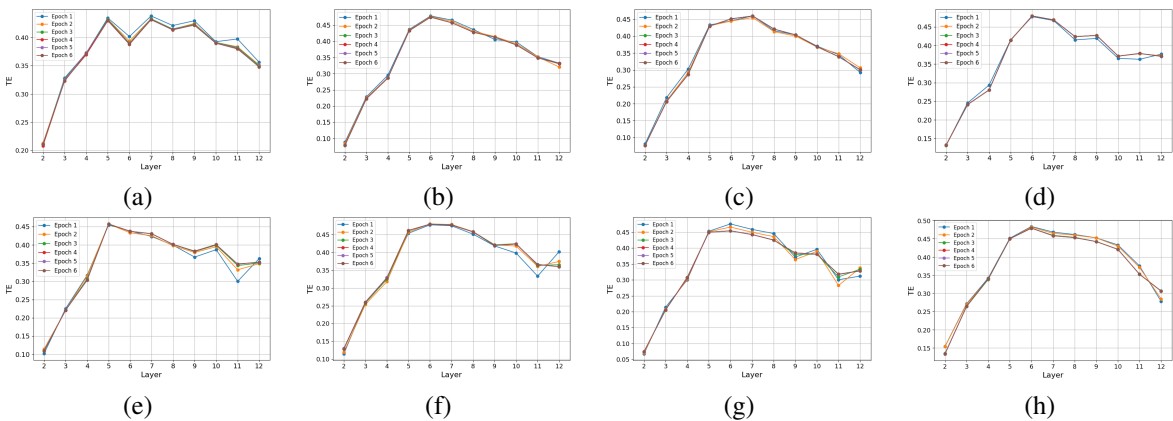

*Figure 13.* **T5-Base (decoder).** TE vs. layer index during 6-epoch full fine-tuning on the GLUE benchmark: (a) MNLI, (b) MRPC, (c) SST-2, (d) CoLA, (e) QNLI, (f) QQP, (g) RTE, (h) STS-B.

to dataset-specific adaptation. Unlike models with sharp mid-depth peaks, T5-Base shows a flatter curve, with only minor decreases in the final layers—evidence that representational flexibility is distributed more evenly across the encoder while the upper layers retain greater adaptive capacity.

In Figure 15b, we show the mean and standard deviation of the TE in the T5-Base decoder, again aggregated over six epochs and twelve datasets. The decoder exhibits a distinct dynamic: the mean starts low ($\approx 0.22$), rises rapidly through lower/middle layers to a peak of $\approx 0.91$ at layer 5, then gradually declines to about $0.67$ in the final layer. This pattern suggests that the early and middle decoder layers introduce and maintain the richest representational transformations, whereas upper layers consolidate and refine representations for task-specific outputs. The standard deviation is largest in the initial layer ($\approx 0.08$), reflecting high cross-dataset variability when forming initial representations, then decreases and stabilizes in the middle layers before rising again toward the output. Altogether, this TE profile underscores the specialization of decoder segments for transformation and output integration during sequence generation.

### H.5.2. INTERPRETATION OF MEAN AND STANDARD DEVIATION OF TE

Analyzing TE profiles across layers provides insight into how architectural design and dataset-specific adaptation shape information dynamics in LLMs. We interpret the *mean* TE at layer $\ell$ as reflecting the architecture's intrinsic propensity for representational *reuse or duplication* at that depth (e.g., driven by depth, attention topology, and residual connectivity). Because the mean is aggregated over datasets and epochs, it captures a stable, architecture-driven pattern of where the model tends to compress, relay, or re-express information versus where it tends to introduce new transformations. In this view, *low* mean TE suggests layers that contribute more novel, task-relevant change, whereas *high* mean TE indicates layers whose

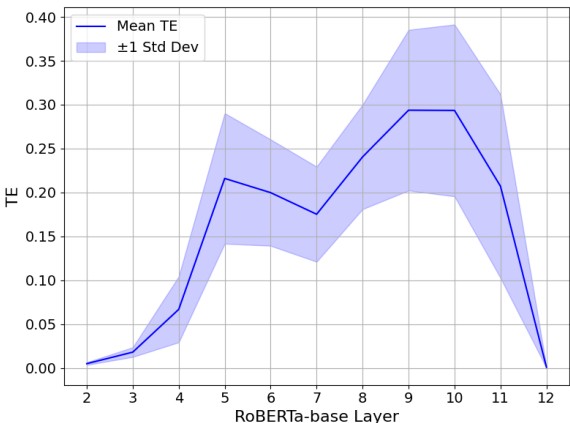

*Figure 14.* **RoBERTa_base (encoder).** Mean and standard deviation of the TE across 12 layers with 6-epoch fine-tuning on 12 datasets.

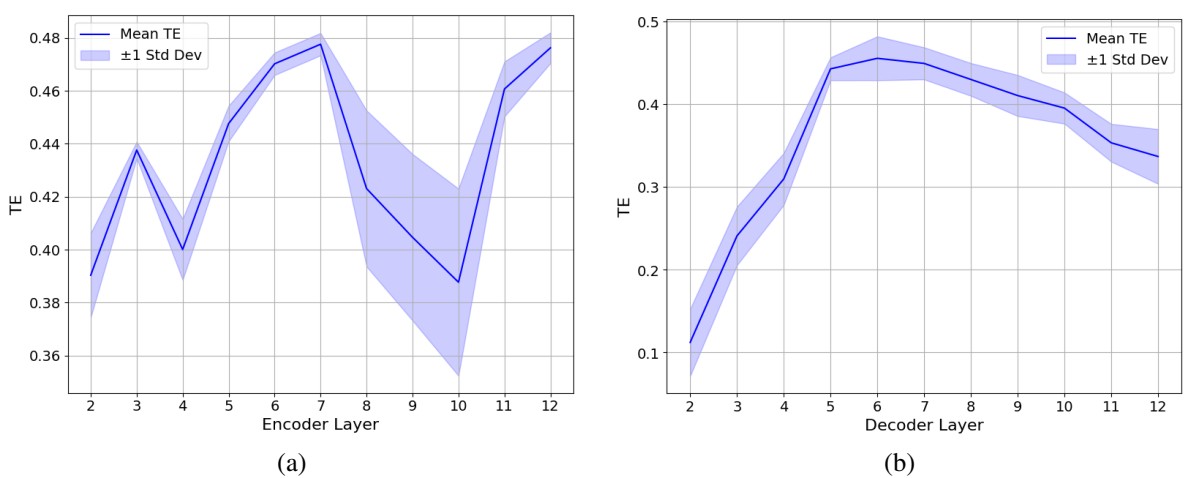

*Figure 15.* **T5-Base.** Mean and standard deviation of the TE with 6-epoch fine-tuning on 12 datasets. (a) Encoder, (b) Decoder.

outputs are more predictable from neighboring layers.

In contrast, the *std* of the TE quantifies how strongly a layer's behavior is modulated by dataset-specific factors. High std marks *adaptation hotspots* whose representational redundancy varies with domain shift, label structure, or semantic variability; low std of TE indicates a more data-agnostic regime where behavior is governed primarily by architectural priors rather than input diversity. Thus, std of TE highlights the layers most responsive to fine-tuning, where the model flexibly adjusts redundancy in service of task demands.

Taken together, this dual perspective—*mean* as an architectural fingerprint and *std* as dataset-induced plasticity—enables a nuanced dissection of functional roles across depth. Practically, layers with *high mean* and *low std* TE are promising candidates for structured pruning or parameter sharing, while layers with *low mean* and/or *high std* warrant preservation or targeted adaptation (e.g., adapters/LoRA), informing both analysis and design of efficient LLMs.

## H.6. Information Flow in Larger LLMs

We evaluate two open-weight decoder-only Transformer LLMs, Llama-3.2-3B and Qwen-2.5-7B. Llama-3.2-3B (Dubey et al., 2024) is a 3B-parameter autoregressive model with 28 Transformer decoder layers, designed for efficient reasoning and long-context inference. Qwen-2.5-7B (Qwen Team, 2024) is a larger 7B-parameter decoder-only model comprising 28 layers, trained on a diverse multilingual and code-rich corpus to provide strong general-purpose performance.

We evaluate Llama-3.2-3B on the GLUE benchmark (8 datasets) (Wang et al., 2018) using a two-stage procedure that

combines LoRA fine-tuning with TE-based layer analysis. Each dataset is tokenized with the Llama tokenizer and padded to a maximum sequence length of 256. The base model is initialized for sequence classification and augmented with rank-4 LoRA on all attention and MLP projection matrices. We fine-tune for two epochs using AdamW with linear warmup. To measure information flow across depth, we register lightweight forward hooks on consecutive decoder blocks and compute a cosine-similarity TE proxy that captures the alignment between successive activation changes across training batches. After each epoch, we obtain a 27-edge TE profile (Layers 2–28), indicating where the model concentrates task-specific computation.

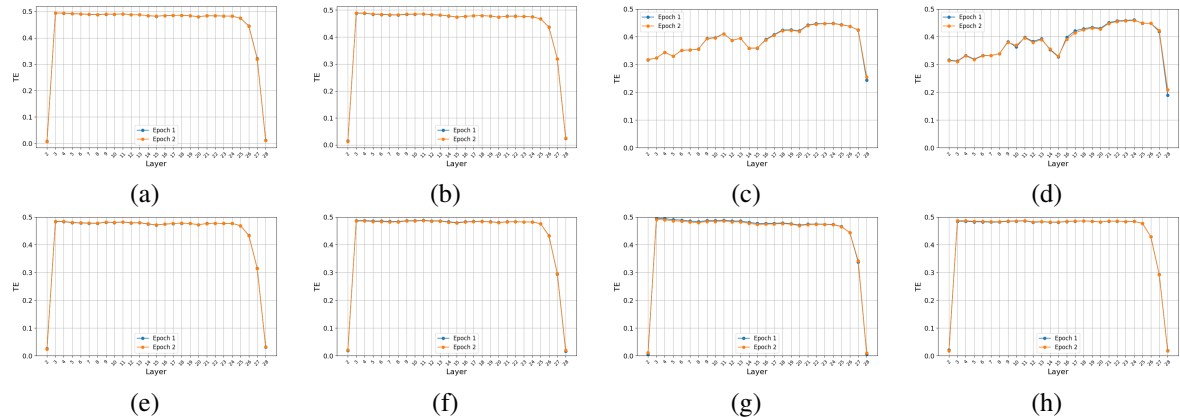

Figure 16. Information flow in **Llama-3.2-3B**. TE vs. layer index during 2-epoch LoRA (rank 4) fine-tuning on the GLUE benchmark: (a) MNLI, (b) MRPC, (c) SST-2, (d) CoLA, (e) QNLI, (f) QQP, (g) RTE, (h) STS-B.

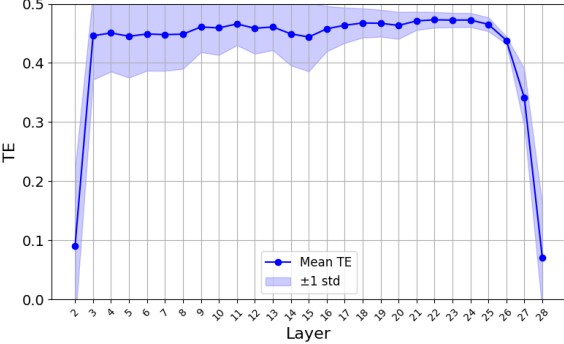

Figure 17. Statistics of information flow in **Llama-3.2-3B.** Mean and standard deviation of the TE across 28 layers with 2-epoch LoRA (rank 4) fine-tuning on 8 datasets in GLUE benchmark.

For Llama-3.2-3B, the resulting TE curves (Fig. 16) exhibit a clear three-zone structure: TE is near zero in the first decoder block, rises rapidly to a plateau around $\approx 0.45$ across most middle layers, and then drops sharply in the final layers. The close overlap between epochs and the small cross-task variance (Fig. 17) show that short LoRA fine-tuning preserves this pattern, with a stable band of mid-level layers carrying the bulk of task-dependent computation while boundary layers behave more like input/output adapters.

We apply the same LoRA fine-tuning and TE analysis pipeline to Qwen-2.5-7B across all GLUE tasks. The model is trained for two epochs with rank-4 LoRA using the Qwen tokenizer and identical optimization settings. After each epoch, we compute layer-wise TE using the same hook-based cosine-similarity metric, yielding a 27-edge TE profile (Layers 2–28) that reveals how Qwen-2.5-7B distributes computation during adaptation.

For Qwen-2.5-7B, the TE curves (Fig. 18) follow a broadly similar shape but with richer internal structure. TE rises steeply in early layers, exhibits a pronounced dip from Layer 3 to Layer 4 across all tasks, then recovers to a stable high plateau ($\approx 0.40$–$0.48$) before decreasing near the model top. The mean and variance summary (Fig. 19) shows that this early dip and top-layer decline are the main sources of cross-task variability, while the mid-depth plateau forms a robust high-TE

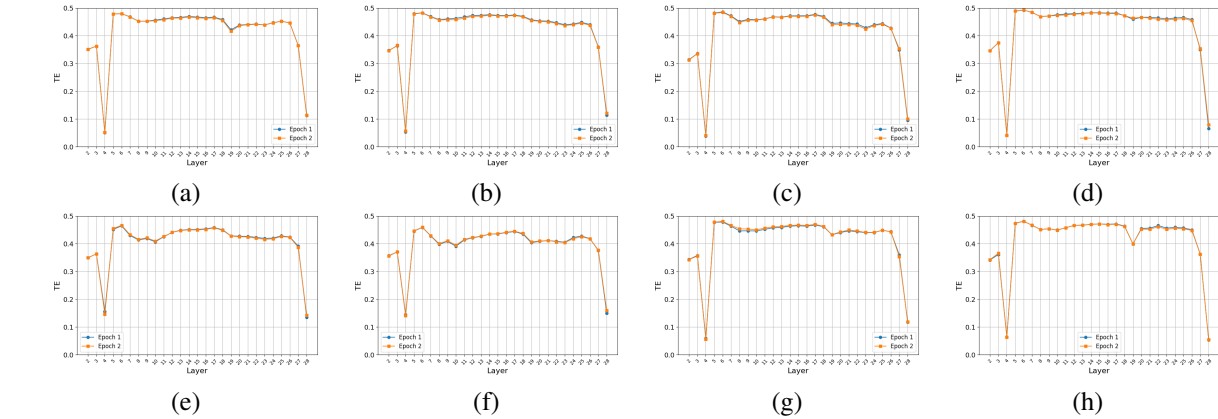

*Figure 18.* Information flow in **Qwen-2.5-7B.** TE vs. layer index during 2-epoch LoRA (rank 4) fine-tuning on the GLUE benchmark: (a) MNLI, (b) MRPC, (c) SST-2, (d) CoLA, (e) QNLI, (f) QQP, (g) RTE, (h) STS-B.

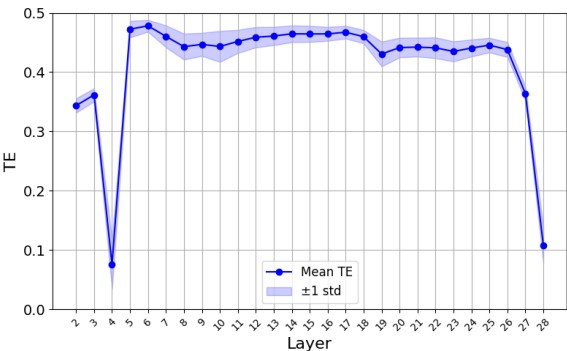

*Figure 19.* Statistics of information flow in **Qwen-2.5-7B.** Mean and standard deviation of the TE across 28 layers with 2-epoch LoRA (rank 4) fine-tuning on 8 datasets in GLUE benchmark.

backbone that dominates computation during GLUE fine-tuning.

## I. Additional Experimental Results for VLMs

### I.1. Information Flow in LLaVA-7B

We study the internal information flow dynamics of LLaVA-1.5-7B (Liu et al., 2023)(Liu et al., 2024) on two image-caption datasets: MSCOCO-2014 and Flickr8k (Marco et al., 2023). LLaVA (Large Language and Vision Assistant) is a multimodal instruction-following model that integrates a CLIP vision encoder with a LLaMA language decoder. In LLaVA-1.5-7B, the vision tower is a CLIP ViT-L/14 backbone (pure encoder model with approximately 0.4B parameters) that encodes the input image into a sequence of visual tokens. LLaVA's text tower is a LLaMA-7B transformer decoder (approximately 7B parameters) responsible for generating natural language.

In LLaVA, a small projection module maps CLIP ViT-L/14's visual embeddings into LLaMA's token embedding space, allowing them to be inserted as special image tokens at the beginning of the decoder input sequence. Because these image tokens participate in the same self-attention mechanism as the textual tokens, visual representations influence the hidden states of progressively deeper decoder layers. This induces a clear direction of information flow from vision to text, enabling the decoder to ground linguistic predictions in the image content. Perturbations to the input image therefore propagate through the transformer stack, producing measurable changes in hidden-state activations across layers.

Our goal is to quantify how visual information propagates through (i) the CLIP vision tower, (ii) the LLaMA-7B language model, and (iii) the cross-modal pathways where visual features condition language generation. To explicitly quantify multimodal interaction, we compute a cross-modal TE proxy from visual tokens to text tokens within the language tower.

For each LLaMA layer $\ell$, we separate hidden states into image tokens and text tokens using the special image token ID, pool each group separately, and form

$$\Delta_\ell^{\text{vis}} = \mathbf{z}_{\ell,\text{vis}}^{(2)} - \mathbf{z}_{\ell,\text{vis}}^{(1)}, \qquad \Delta_\ell^{\text{text}} = \mathbf{z}_{\ell,\text{text}}^{(2)} - \mathbf{z}_{\ell,\text{text}}^{(1)}. \tag{74}$$

We then define a vision-to-text TE score per language layer as

$$\text{TE}_\ell^{\text{vis}\rightarrow\text{text}} \approx \tfrac{1}{2}\mathbb{E}\big[\cos^2(\Delta_\ell^{\text{vis}}, \Delta_\ell^{\text{text}})\big]. \tag{75}$$

This measures how strongly changes in the aggregated visual representation are aligned with changes in the aggregated text representation at the same depth.

In Fig. 20, we plot the layer-wise TE across 5 epochs of LoRA (rank 4) fine-tuning on the Flickr8k dataset. The three panels respectively show the TE progression in the CLIP ViT-L/14 vision tower, the LLaMA-7B language tower, and the cross-modal vision→text pathway. In Fig. 21, we summarize the corresponding TE curves when the same fine-tuning procedure is applied to the larger MSCOCO dataset. The overall shapes of the TE trajectories are consistent with Flickr8k, indicating that LLaVA exhibits a stable information-flow hierarchy across datasets. Finally, in Fig. 22, we present the mean and standard deviation of the TE across all 10 experiments (5 epochs × 2 datasets), providing a consolidated view of how information propagates through the vision tower, the language tower, and the cross-modal fusion layers. These averaged curves reveal highly consistent structural patterns, with most variance localized in mid-layer dynamics of the language tower and in the deeper layers of the vision→text pathway.

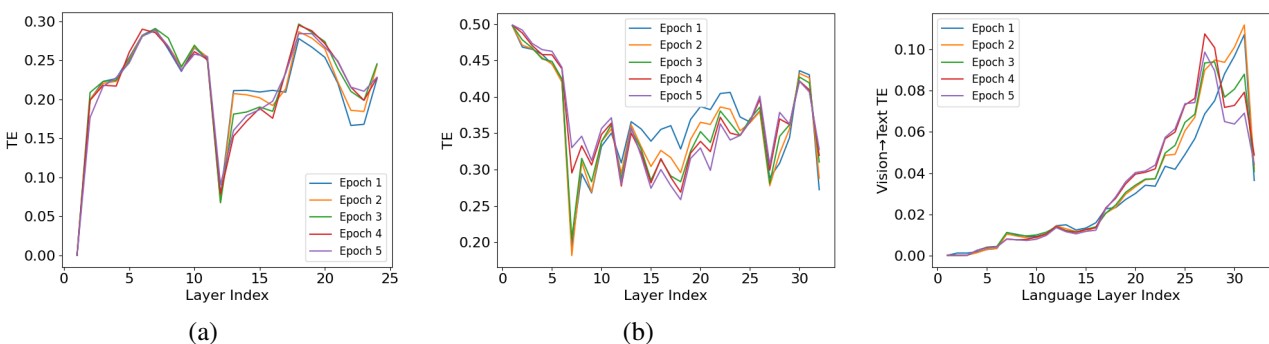

(a)          (b)

*Figure 20.* Information flow in **LLaVA-1.5-7B** using Flick8k. TE vs. layer index during 5-epoch LoRA (rank 4) fine-tuning: (a) vision tower, (b) text tower, (c) vision→text.

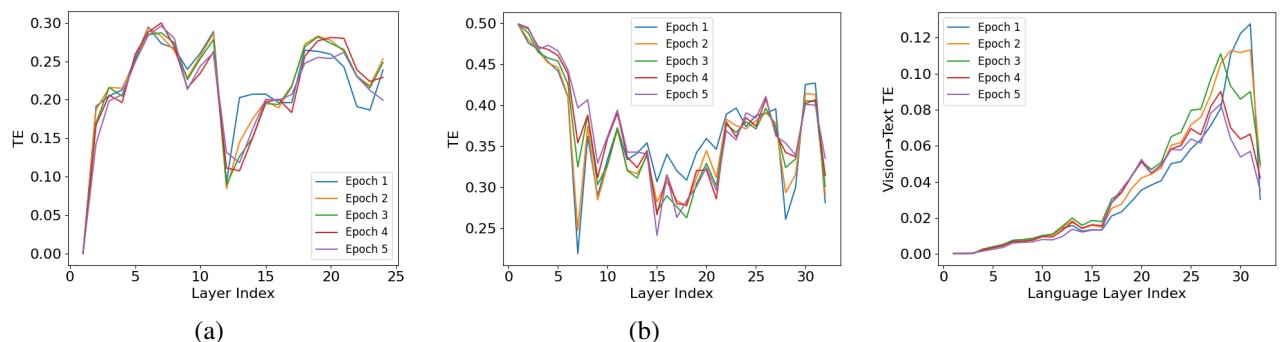

(a)          (b)

*Figure 21.* Information flow in **LLaVA-1.5-7B** using MSCOCO. TE vs. layer index during 5-epoch LoRA (rank 4) fine-tuning: (a) vision tower, (b) text tower, (c) vision→text.

Across both Flickr8k and MSCOCO, we observe a highly consistent hierarchy of information flow within LLaVA-1.5-7B during 5-epoch LoRA (rank 4) fine-tuning.

1. *Vision tower.* The CLIP ViT-L/14 vision encoder exhibits a stable TE profile across epochs and datasets, with characteristic peaks in mid-to-late layers corresponding to the progression from low-level patch embeddings to higher-level semantic abstractions. The TE curves remain largely unchanged across training epochs, indicating that LoRA

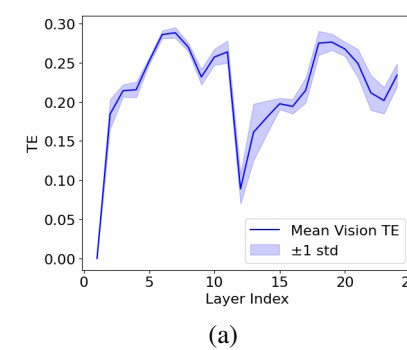 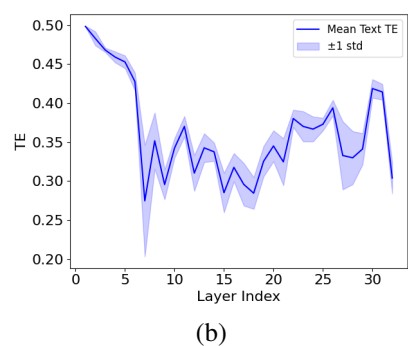 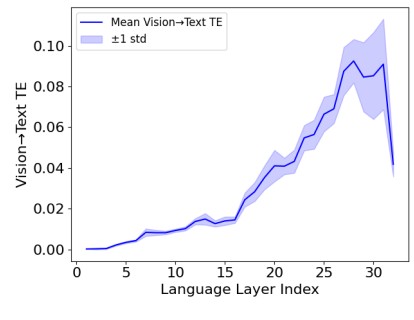

(a)  (b)

*Figure 22.* Statistics of information flow in **LLaVA-1.5-7B**. Mean and standard deviation of the TE across 5-epoch LoRA (rank 4) fine-tuning on 2 datasets (MSCOCO and Flick8k). (a) vision tower, (b) text tower, (c) vision→text.

updates in the language tower do not propagate back into the frozen vision backbone. The similarity between MSCOCO and Flickr8k further demonstrates that the vision tower's information propagation is dataset-invariant and robust to scale.

2. *Language tower.* The LLaMA-7B decoder shows consistently high TE values (approximately 0.30–0.50) across all layers and all training epochs. This is consistent with deep autoregressive transformers, whose representations are iteratively refined layer by layer. Although mid-layer variance is higher on the smaller Flickr8k dataset, the overall TE structure remains stable across datasets. LoRA fine-tuning slightly increases TE magnitudes in deeper layers, suggesting stronger intra-language conditioning while leaving the qualitative information-flow topology intact.

3. *Cross-modal vision→text flow.* The most distinctive result is the persistent upward trend in cross-modal TE across depth. Early LLaMA layers display near-zero visual influence, implying minimal integration of image features. TE then increases steadily with layer depth, reaching its maximum in the top decoder blocks. This indicates that LLaVA performs multimodal fusion primarily in the upper layers of the language tower, where visual perturbations have the strongest effect on textual hidden states. This trend becomes sharper across epochs, especially for Flickr8k, suggesting that small datasets encourage the model to localize visual grounding into a narrower set of upper layers.

Taken together, the results show that LLaVA-7B maintains a stable and interpretable three-part information-flow structure across datasets and training conditions: (i) a dataset-invariant vision encoder with moderate TE in deeper blocks, (ii) a strong and persistent language-refinement process throughout the decoder, and (iii) a hierarchical cross-modal pathway in which visual information is injected primarily in the upper language layers. LoRA fine-tuning modulates the *strength* of these flows but preserves their overall topology, indicating that multimodal coupling adapts while the underlying structural roles of visual, textual, and cross-modal layers remain unchanged.

### I.2. Information Flow among Attention Heads in CLIP

We extend information flow analysis to *attention heads*. The aim is interpretability: to expose which heads aggregate information into the special token (CLS in vision; EOT in text), which spread information out from it, and which largely pass information without strong directional effect.

**Per–head CLS/EOT mass.** For layer $\ell \in \{0, \ldots, L-1\}$ and head $h \in \{1, \ldots, H\}$, let $P^{(\ell,h)} \in [0,1]^{T \times T}$ be attention probabilities (rows: queries, columns: keys) and let $c$ index the special token (CLS/EOT). We summarize each head by its average attention *into* and *out of* the special token:

$$m_{\text{in}}^{(\ell,h)} := \mathbb{E}_q\left[P_{q \to c}^{(\ell,h)}\right], \qquad m_{\text{out}}^{(\ell,h)} := \mathbb{E}_k\left[P_{c \to k}^{(\ell,h)}\right], \tag{76}$$

where expectations are over tokens (and minibatch). Large $m_{\text{in}}$ indicates funneling many queries into CLS/EOT (sink–like), whereas large $m_{\text{out}}$ indicates broadcasting from CLS/EOT.

**Layerwise TE–aware head labeling.** Within each layer we standardize the per–head CLS/EOT attention masses $m_{\text{in}}^{(\ell,h)}$ and $m_{\text{out}}^{(\ell,h)}$ across heads using $z$–scores,

$$\tilde{m}_{\text{in}}^{(\ell,h)} = \text{zscore}\big(m_{\text{in}}^{(\ell,h)}\big), \qquad \tilde{m}_{\text{out}}^{(\ell,h)} = \text{zscore}\big(m_{\text{out}}^{(\ell,h)}\big),$$

and use the $\text{TE}_{\ell,h}$ as an impact proxy. Let $\text{pct}_p(\cdot)$ denote the $p$–th percentile computed *over heads in the same layer*. We form layerwise thresholds

$$\tau_{\text{in}} = \text{pct}_{p_{\text{in}}}\big(\tilde{m}_{\text{in}}^{(\ell,\cdot)}\big), \quad \tau_{\text{out}} = \text{pct}_{p_{\text{out}}}\big(\tilde{m}_{\text{out}}^{(\ell,\cdot)}\big), \quad \tau_{\text{TE}}^{\text{lo}} = \text{pct}_{p_{\text{TE}}^{\text{lo}}}\big(\text{TE}_{\ell,\cdot}\big), \quad \tau_{\text{TE}}^{\text{hi}} = \text{pct}_{p_{\text{TE}}^{\text{hi}}}\big(\text{TE}_{\ell,\cdot}\big).$$

In our experiments we use $p_{\text{in}}{=}55$, $p_{\text{out}}{=}80$, $p_{\text{TE}}^{\text{lo}}{=}25$, and $p_{\text{TE}}^{\text{hi}}{=}60$, and we *require* sufficient TE to confirm a non–benign role.

**Decision rules (per head $h$ in layer $\ell$).**

$$\textbf{Broadcaster: } \Big(\tilde{m}_{\text{out}}^{(\ell,h)} \geq \tau_{\text{out}} \ \wedge \ \tilde{m}_{\text{in}}^{(\ell,h)} < \tau_{\text{in}}\Big) \ \wedge \ \Big(\text{TE}_{\ell,h} \geq \tau_{\text{TE}}^{\text{lo}}\Big), \tag{77}$$

$$\textbf{Sink: } \Big(\tilde{m}_{\text{in}}^{(\ell,h)} \geq \tau_{\text{in}} \ \wedge \ \tilde{m}_{\text{out}}^{(\ell,h)} < \tau_{\text{out}}\Big) \ \wedge \ \Big(\text{TE}_{\ell,h} \geq \tau_{\text{TE}}^{\text{lo}}\Big), \tag{78}$$

$$\textbf{Ambiguous: } \text{otherwise.} \tag{79}$$

Ambiguous heads are resolved using TE as a tie–breaker: if $\text{TE}_{\ell,h} \geq \tau_{\text{TE}}^{\text{hi}}$, assign

$$\text{role}(h) = \begin{cases} \text{sink,} & \text{if } \tilde{m}_{\text{in}}^{(\ell,h)} \geq \tilde{m}_{\text{out}}^{(\ell,h)}, \\ \text{broadcaster,} & \text{otherwise,} \end{cases}$$

else assign *benign*. Finally, as a safety demotion, any head provisionally labeled sink/broadcaster with $\text{TE}_{\ell,h} < \tau_{\text{TE}}^{\text{lo}}$ is relabeled *benign*. This scheme keeps labels driven primarily by CLS/EOT mass patterns while using TE to (i) require minimal functional impact for non–benign roles and (ii) stabilize borderline cases.

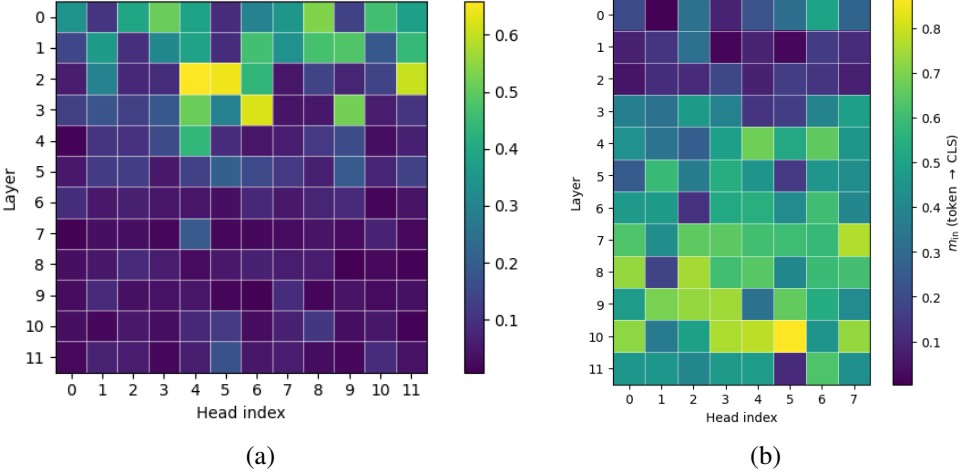

(a)          (b)

*Figure 23.* $m_{in}$ of each head in every layer for CLIP ViT-B/16. (a) Vision tower. (b) Text tower.

We trained CLIP ViT-B/16 with LoRA injection using rank $r{=}8$, scaling factor $\alpha{=}16$, and no dropout. The dataset was MSCOCO-2014. Optimization was performed with AdamW at a learning rate of $10^{-4}$ and weight decay 0.0. The classification was done after two epochs of training. Heatmaps of $m_{\text{in}}$ versus head index (layers on $y$–axis, heads on $x$–axis) was plotted in Fig. 23. They reveal that, in the *vision* tower, only a few heads exhibit strongly elevated $m_{\text{in}}$ in early–to–mid layers, and this concentration softens in deeper layers—consistent with CLS being a modest collector while spatial interactions dominate. In contrast, the *text* tower shows steadily rising $m_{\text{in}}$ across several heads with depth, indicating strong collection into EOT. Per–head TE heatmaps in Fig. 24 complement this view: vision contains sporadic,

high–impact heads (isolated bright cells), whereas text displays more uniformly elevated TE in later layers, suggesting broader participation in the layer's net update. Applying the rules above yields head–role grids in Fig. 25 showing that vision layers retain a balanced mix with many benign heads interleaved with pockets of sinks/broadcasters, while text layers polarize more strongly by depth, with benign heads shrinking and clearer sink/broadcaster structure emerging.

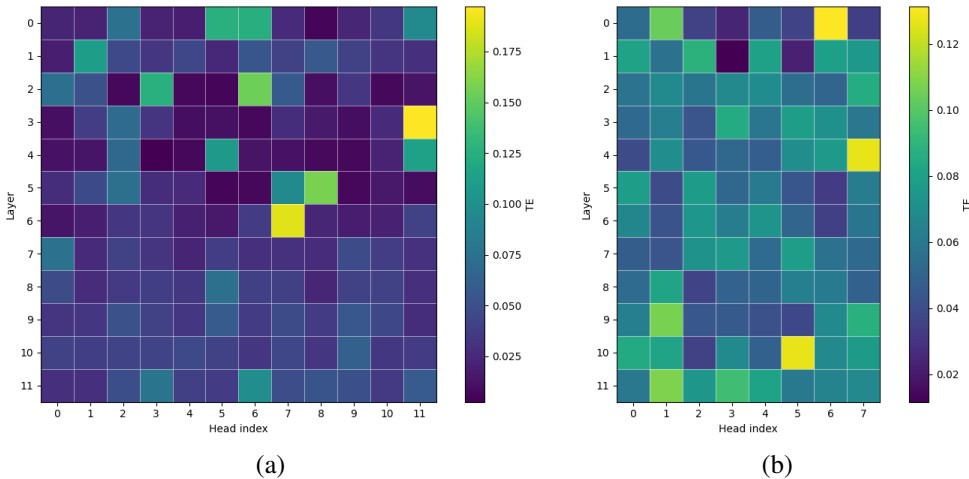

(a)  (b)

*Figure 24.* TE of each attention head in every layer. (a) Vision tower, (b) text tower.

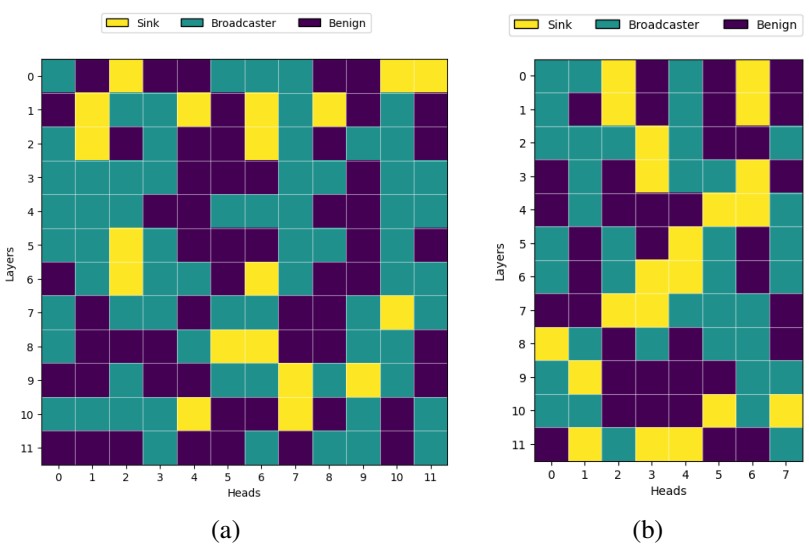

(a)  (b)

*Figure 25.* Attention heads classification in each layer for CLIP ViT-B/16. (a) Vision tower. (b) Text tower.

Head–level roles provide compact, layer–wise *maps of responsibility*. They support: (i) attribution of cross–modal interactions (which heads write into vs. read from CLS/EOT); (ii) diagnosis of training pathologies (e.g., layers dominated by sinks with little broadcasting); (iii) targeted qualitative analysis (visualizing only broadcaster heads when explaining an image–text match); and (iv) gentle interventions such as role–aware temperature scaling or dropout targeted at persistently benign heads to improve robustness—without changing model capacity. Compared to raw attention, the joint view of $m_{\text{in}}/m_{\text{out}}$ with TE separates *where probability mass goes* from *which heads actually drive the layer update*, yielding a clearer interpretation of the communication structure inside CLIP's vision and text towers. Role-aware analysis turns attention heads into readable components: *sinks* aggregate evidence into CLS/EOT, *broadcasters* disseminate global context from it, and *benign* heads provide supportive routing. Visual and quantitative summaries make these behaviors observable, and they unlock practical levers for debugging, steering, and transferring CLIP.

## J. Computing Infrastructure, Datasets, and Runtime

In LLMs, RoBERTa-base, T5-base, Llama-3.2-3B, and Qwen-2.5-7B are all pretrained models. For fine-tuning using GLUE and CoT datasets, we use a subset of each of the 12 datasets. Specifically, we randomly select up to 5,000 training examples per dataset (using all available data if the training set contains fewer than 5,000 samples). For evaluation, we randomly sample 1,000 examples from each test set.

In VLMs models, both CLIP ViT-B/16 and LLaVA-7B are pre-trained. MSCOCO-2014 (Lin et al., 2014) was used for both models. The MSCOCO-2014 dataset consists of 82,783 training images and 40,504 validation images. The dataset is organized into two image folders and accompanying annotation files, which provide the class labels and bounding box locations for objects contained in each image. In addition to object detection and segmentation, each image is paired with five human-written captions, making MSCOCO a widely used benchmark for image–text tasks such as retrieval and captioning. In LLaVA-7B experiment, Flick8k was also used. Flickr8k (Marco et al., 2023) is a widely used benchmark for vision-language modeling and captioning. It contains 8,000 natural images collected from Flickr, each annotated with five human-written captions that describe salient objects, actions, and scene context. Despite its modest size, Flickr8k captures diverse real-world scenarios and provides a clean testbed for analyzing image–text alignment, cross-modal information flow, and multimodal fusion dynamics in VLMs.

Table 6 summarizes the runtimes of key experiments conducted on a Colab A100 GPU. These results highlight that both the choice of model and the specific method have a significant impact on runtime.

| Experiments | Runtime |
|---|---|
| RoBERTa-base using GLUE datasets | 2h 50m |
| RoBERTa-base using CoT datasets | 1h 30m |
| T5-base using GLUE Datasets | 3h 20m |
| T5-base using CoT datasets | 1h 50m |
| KE for RoBERTa using GLUE datasets | 2h 35m |
| KE for T5 using CoT datasets | 1h 20m |
| LayerDropout for RoBERTa using GLUE | 1h 50m |
| LayerDropout for T5-base using CoT | 1h 10m |
| LayerDropout for RoBERTa using GLUE | 1h 50m |
| LayerDropout for T5-base using CoT | 1h 10m |
| SlimLLM for RoBERTa using GLUE datasets | 2h 5m |
| SlimLLM for T5 using CoT datasets | 1h 12m |
| RoBERTa-base FT with $\alpha = 0.02$ | 2h 50m |
| Llama-3.2-3B information flow using GLUE | 3h 25m |
| Qwen-2.5-7B information flow using GLUE | 4h 35m |
| LLaVA-1.5-7B using MSCOCO | 5h 20m |
| LLaVA-1.5-7B using Flick8k | 3h 10m |
| CLIP ViT-B/16: TE Layer Pruning | 6h 20m |
| CLIP ViT-B/16: Streamline Layer Pruning | 6h 20m |
| CLIP ViT-B/16: Short-LVLM Layer Pruning | 6h 35m |
| CLIP ViT-B/16: Attention head classification | 3h 35m |

*Table 6.* Runtime details for key experiments.

## K. Limitations

Although transfer entropy provides a principled and interpretable measure of directed information flow, our approach has several limitations. Conceptually, exact TE requires estimating conditional densities over very large hidden states, which is infeasible for modern LMs and VLMs. Our approach therefore relies on (i) a local linear-Gaussian approximation via

Jacobians (Theorem 4.1) and (ii) a further rank-one approximation that leads to the cosine-similarity estimator (Theorem 4.3). These approximations introduce several potential sources of bias: for example, finite-batch effects and local nonlinearity can lead to noisy or slightly inflated TE values when random fluctuations align across layers, while the rank-one assumption can underestimate TE when information propagates through multiple independent directions.

Practically, we mitigate these issues by (a) averaging the estimator over many mini-batches and training steps, (b) working with pooled representations (rather than individual tokens) to reduce variance, and (c) focusing on relative patterns (shape of TE vs. depth, changes across epochs, and comparisons across architectures) rather than interpreting absolute TE values as exact information in bits. We have added a dedicated paragraph on these limitations in Section 4.2 and a brief remark in the Discussion, emphasizing that our TE curves should be viewed as stable, qualitative probes of layer-to-layer information flow rather than precise, fully nonparametric TE estimates. We also point readers to Appendix G, where we compare the cosine-based approximation to exact TE in a controlled low-dimensional setting and observe close agreement in the regimes most relevant to our experiments.

## L. Use of Large Language Models

During the preparation of this paper, we used ChatGPT 5 to assist with writing and experimentation. On the writing side, it helped with grammar and spelling checks, and polished several paragraphs. For experiments, it helped diagnose and fix bugs in our Python implementation.

