# OpenReview forum: "Transfer Entropy as a Measure of Information Flow in LLMs and VLMs"
_ICML.cc/2026/Conference — Submitted to ICML 2026_

### Official Review · Reviewer_BE5Q · 2026-02-28

**Soundness:** 4
**Presentation:** 3
**Significance:** 3
**Originality:** 3
**Overall Recommendation:** 5
**Confidence:** 3

**Summary:**

This paper addresses the analysis of information transfer in LLMs and VLMs, by introducing the Transfer Entropy  metric. Through solid theoretical derivation of a linear-complexity approximate estimator, the study characterizes TE structures across diverse architectures. It establishes a foundation for model interpretability and compression. Overall, the work presents a novel perspective, supported by rigorous theoretical foundation and comprehensive empirical results.

**Compliance With Llm Reviewing Policy:**

Affirmed.

**Final Justification:**

The authors provide comprehensive experiments and analysi and my concerns have been fully addressed. I agree with the significance and completeness of their work, with solid mathmatical analysis and extensive experiments. I would like to increase my score to 5.

**Key Questions For Authors:**

1. The formulation in lines 270-281 appears confusing. TE typically measures the ability of a preceding layer's variations to predict a succeeding layer's future state (namlely, the gain in mutual information), where higher TE implies higher predictive certainty. However, the text here interprets high-TE regions as exhibiting high "uncertainty and flexibility," while interpreting low-TE output layers as having high certainty. Does this appear to have a conceptual contradiction in your interpretation?
2. Regarding the VLM analysis in Appendix I.1, "vision→text TE" is calculated using the cosine similarity of variations between visual and textual tokens. Given that visual and textual representations are in different embedding spaces, are their scales and semantics directly comparable? Could high similarity merely stem from co-fluctuations induced during training? Would a control experiment involving noise-injected text or images be necessary to validate the significance of these correlations?

**Strengths And Weaknesses:**

**Strengths:**
1. Novel methodology: The paper innovatively applies Transfer Entropy to analyze information flow within Transformers, providing valuable guidance for downstream applications such as interpretability and pruning.
2. Rigorous theoretical derivation: The authors derive the TE estimator through rigorous mathematical formulation, establishing a solid theoretical foundation for the proposed metric.
3. Comprehensive empirical results: The effectiveness of the method is demonstrated through extensive experiments, including TE analysis across diverse architectures and comparative pruning studies, yielding profound insights.

**Weaknesses:**
1. Lack of efficiency analysis: While the paper theoretically derives a linear-complexity approximation algorithm, it lacks a concrete experimental evaluation or discussion regarding actual computational efficiency like latency.
2. Insufficient discussion on fine-tuning epochs: The pruned models undergo fine-tuning for only 3 or 5 epochs. How the number of epochs influences model performance is unclear. Specifically, could increasing the number of epochs narrow the performance gap between baselines and the proposed method, or potentially allow baselines to surpass it?

---

> ### Author Rebuttal · Authors · 2026-03-29
>
> We appreciate the reviewer’s careful reading and insightful comments. We respond to each point below.
>
> **W1** We would like to clarify that the paper already includes a theoretical efficiency discussion in Appendix F, as well as runtime measurements for the main experiments on a Colab A100 GPU in Appendix J (Table 6). As a concrete additional example, we benchmarked RoBERTa on MNLI before and after TE-based pruning under the same batch size and sequence length. Pruning 4 layers reduced latency from 46.50 ms/batch to 37.14 ms/batch (batch size = 32), giving a 1.25$\times$ speedup and a 25.2% throughput gain (688.1 $\rightarrow$ 861.5 examples/s). Peak memory also decreased from 3213.8 MB to 3063.2 MB. We further measured the TE scoring overhead during selection: forward+TE took 49.29 ms/batch versus 46.56 ms/batch for forward only, i.e., an overhead of 2.72 ms/batch (5.85%). Since this cost is incurred only once during pruning/selection and not at deployment, the pruned model still yields a net inference-time efficiency benefit.
>
> **W2** To clarify the effect of recovery epochs, we additionally compare post-pruning fine-tuning for 3 vs. 5 epochs on RoBERTa-base for our TE pruning and the strongest baseline, SlimLLM. As shown in the table, both methods reach near-converged performance quickly, and extending recovery from 3 to 5 epochs yields only small improvements. Importantly, the relative ranking does not change: TE pruning remains consistently better than SlimLLM across all tasks at both 3 and 5 epochs. This suggests that the performance gap is not an artifact of too few recovery epochs, and that giving more epochs does not allow the baseline to catch up in this setting.
>
> | Method         | Epochs | MNLI   | MRPC   | SST-2  | CoLA   | QNLI   | QQP    | RTE    | STS-B  |
> |----------------|-------:|-------:|-------:|-------:|-------:|-------:|-------:|-------:|-------:|
> | TE pruning     | 3      | 0.7820 | 0.8407 | 0.9174 | 0.8188 | **0.8552** | 0.8359 | 0.7256 | 0.9054 |
> | TE pruning | 5  | **0.7860** | **0.8480** | **0.9186** | **0.8255** | 0.8548 | **0.8361** | **0.7437** | **0.9063** |
> | SlimLLM        | 3      | 0.7630 | 0.8039 | 0.8979 | 0.8082 | 0.8545 | 0.8341 | 0.6787 | 0.9041 |
> | SlimLLM        | 5      | 0.7660 | 0.8088 | 0.8979 | 0.8092 | 0.8548 | 0.8359 | 0.6787 | 0.9043 |
>
> **Q1** Our discussion in lines 271–278 is intended from a feature-learning perspective rather than as the formal definition of TE. Specifically, low TE in the input layers suggests relatively stable feature extraction, while elevated TE in the middle layers indicates that representations are still being actively reshaped and integrated, i.e., a more flexible stage of feature learning. By contrast, low TE in the output layers is consistent with features becoming more stable and finalized for task-aligned decoding.  This does not contradict our pruning interpretation in lines 282–295: high TE indicates that a layer’s update is more predictable from neighboring layers and is therefore more redundant/prunable, whereas low-TE layers are more stable and indispensable.
>
> **Q2** Thank you for this important question. In our Appendix I.1 implementation, the “vision$\rightarrow$text TE” is not computed between raw CLIP visual embeddings and raw text embeddings from different spaces. Instead, for each LLaMA layer, we take the same decoder hidden-state tensor, separate it into image-token and text-token subsets using the image token ID, average each subset, and then compute TE proxy. Thus, the comparison is made within a shared hidden space at the same language layer, not across incomparable raw modality spaces. Moreover, the perturbation is applied only to the image pixels, so any resulting change in text-token states reflects propagation through the model’s cross-modal pathway rather than an arbitrary text-side fluctuation. The consistent upward trend across both Flickr8k and MSCOCO also argues against purely accidental co-fluctuation. We agree that an additional null control (e.g., permuted image-token groups or noise-injected text) would further strengthen the analysis, but we view it as a useful robustness check rather than a prerequisite for the qualitative conclusion.

---

> > ### Author Rebuttal · Reviewer_BE5Q · 2026-04-02
> >
> > Dear Authors,
> >
> > Thank you for the comprehensive experiments and analysis. My concerns have been fully addressed. I agree with the significance and completeness of your work. I would like to increase my score to 5.

---

> > > ### Author Response · Authors · 2026-04-05
> > >
> > > Thank you for your acknowledgement. We greatly appreciate your positive feedback and your decision to raise the score to 5.

---

### Official Review · Reviewer_AXcK · 2026-03-12

**Soundness:** 2
**Presentation:** 1
**Significance:** 2
**Originality:** 2
**Overall Recommendation:** 3
**Confidence:** 3

**Summary:**

This paper proposes using the Transfer Entropy measure of directional information flow to discover interdependencies between model layers in LLMs and VLMs. This method is brought up in contrast to existing methods such as self-attention which can be difficult to interpret and provide results that are not necessarily justified theoretically. Specifically, the authors introduce small gaussian pertubations in embeddings across layers to detect minor changes in activations that can calculate Transfer Entropy empirically with theoretical justification on high-dimensional neural networks. The authors use Transfer Entropy for the task of language model layer pruning by removing layers according to transfer entropy. The proposed metric is also used to determine important layers in transformer based VLMs and LLMs.

**Compliance With Llm Reviewing Policy:**

Affirmed.

**Final Justification:**

This paper proposes a novel metric but does not sufficiently tie it down to behavior in the models it analyzes.

Until my rebuttal the sole analysis of this information metric was for the (as the authors admit) non-primary focus of layer pruning. This has been now extended to token pruning and attention head pruning. However, this is not a pruning paper. This is an information flow paper where pruning is an evaluation for their algorithm.

The newly introduced token flow metrics improve the analysis of information flow but I believe that the paper should better analyze how information flows in a paper about an information flow metric. The results on where TE is high or low (most of the results in the paper) are meaningless until that is accomplished. This requires a large change to the manuscript.

**Key Questions For Authors:**

Are there other settings within your compute budget that you can apply transfer entropy? I like that the approach attempts to be more rigorous than analyzing attention, however it is not evaluated on a significant enough variety of tasks. For example, can transfer entropy be applied between tokens instead of layers? Attention is commonly used as a reference for a non-theoretically motivated and less explanatory technique, however unlike the main paper Attention also allows for inter-token dependence instead of simply layer pruning.

**Limitations:**

The proposed approach is model agnostic and does not seem expensive to train. Limitations are most significant with respect to soundness of the experiments, not the approach itself.

**Strengths And Weaknesses:**

Soundness. Strengths: The transfer entropy technique is established in prior work and seems to be appropriately transferred to LLMs. The proposed method of using small gaussian noise is simple and seems generally theoretically well-founded. The existing results do suggest that transfer entropy at least weakly correlates with useful properties in LLMs, as evidenced by pruning experiments.
Weaknesses: The evaluation of transfer entropy is untested outside of layer pruning in modest-sized networks. Specifically, the primary quantifiable table for LLM evaluation (Table 1) against existing baselines evaluates in an extremely simple setting where exhaustive search is still feasible (removing 4 layers from a 12 layer network). Other existing figures, while appropriately demonstrating what transfer entropy discovers, does not validate the the technique is necessarily meaningful. Table 2 only demonstrates extremely small performance improvements over simple baselines.

Presentation: The presentation is extremely longwinded and makes results tedious to understand. The abstract reads as an extended introduction and summary of contributions. Equations do not flow with the paper, and the bulk of the technique is instead described in the Appendix. Equations are also not fully connected to one another. For example, Equation 1 and Equation 2 are said to be equivalent, however (1) is a pointwise estimate and (2) is a distributional mutual information. Is equation (1) missing an expectation?

Figures are extremely small and difficult to read. On my computer, I need 400% zoom to make out label text, and this holds throughout plots in the work. This paper is not formatted to par with the standards of ICML.

Significance: Strengths: A theoretically justified method of determining information flow in Language Models as opposed to attention analysis is a useful contribution to the community.
Weaknesses: The current evaluations do not justify that this approach works at scale or in important settings. The primary downstream effect of transfer entropy is layer pruning for a 12 layer network, and the baselines the paper compares against are fairly simple. Although this approach may have more significant applications, as the problem being addressed is broadly important, the results are not strong enough to imply that the contribution is significant.

Originality: This paper applies an existing approach, Transfer Entropy, to analyze information flow in language models. The approach of using small Gaussian pertubations is original for measuring Transfer Entropy, and the theory is sufficient in depth for ICML. The evaluations of layerwise importance are not particularly original and do not deepen understanding of existing approaches, where layerwise importance of model layers is well studied under other metrics. This paper is sufficiently original, however the insights provided my the method are not novel or significantly higher quality than previous techniques

---

> ### Author Rebuttal · Authors · 2026-03-30
>
> We thank the reviewer for the thoughtful comments and constructive suggestions. We address the main concerns below.
>
> **W1** In the rebuttal, we have evaluated TE-based layer pruning on **LLaVA-1.5-7B** for **VQAv2**. As shown below, TE layer pruning consistently achieves the best soft accuracy (%) across all pruning levels. Streamline and Short-LVLM are cited in Lines 358-360 of the paper.
>
> | \# Layers Pruned | TE Pruning | Streamline Pruning | Short-LVLM |
> |:--:|:--:|:--:|:--:|
> | 2 | **0.5370** | 0.4498 | 0.4548 |
> | 4 | **0.3798** | 0.2937 | 0.2832 |
> | 6 | **0.2867** | 0.1352 | 0.2248 |
> | 8 | **0.2737** | 0.1220 | 0.1472 |
>
>
> **W2** Eq. (1) is intended as the standard probability-weighted definition of transfer entropy, so the expectation is already implicit in the summation against the joint distribution $p(y_{t+1}, y_t, x_t)$. It is therefore equivalent to the conditional mutual information form in Eq. (2), rather than a pointwise estimate.
>
> **W3** The paper analyzes TE-based information flow across a diverse set of architectures and scales, including RoBERTa, T5, Llama-3.2-3B, Qwen-2.5-7B, CLIP ViT-B/16, and LLaVA-1.5-7B, spanning both LLMs and VLMs. In the submission, we evaluate TE-based layer pruning on RoBERTa, T5, and CLIP ViT-B/16. In the rebuttal, we further broaden the empirical evaluation with TE-based layer pruning on LLaVA-1.5-7B for VQAv2 (**W1**), as well as TE-based token pruning on LLaVA-1.5-7B for VQAv2 and on CLIP for MSCOCO (**Q1**).
>
> **W4**  The originality here is the introduction of a directed, conditional information-flow framework for Transformers, together with a practical perturbation-based TE estimator for high-dimensional hidden states. TE asks whether layer $\ell$ provides new predictive information about the update of layer $\ell+1$, beyond what is already explained by the receiver’s own previous state. This enables analyses that go beyond standard layer-importance studies: the paper uncovers distinct depth-wise computation patterns across RoBERTa, T5, Llama, Qwen, CLIP, and LLaVA, and also supports role-aware head classification in VLMs. We therefore view the main novelty as a new directed-information lens for LLMs and VLMs, rather than only the pruning application.
>
> **Q1** Thank you for this suggestion. We agree that evaluating TE beyond layer pruning is important. We have applied TE to token pruning in VLMs, which directly addresses inter-token dependence.
> Specifically, we score tokens using their incoming and outgoing TE, and our ablations show that tokens with larger $TE_{out} − TE_{in}$ are the best pruning candidates. Within our compute budget, we evaluated this on CLIP ViT-B/16 on MSCOCO and LLaVA-1.5-7B on VQAv2, comparing against attention/importance-based baselines including SmartTrim [1], Top-ViM [2], and SCOPE [3].
>
> **Table:** CLIP ViT-B/16 on MSCOCO retrieval under joint vision and text token pruning. Comparison of our TE-based pruning with SmartTrim [1] and Top-ViM [2] when pruning the same percentage of tokens for each layer in both the vision and text towers.
>
> | Pruning | Method | I→T R@1 | I→T R@5 | I→T R@10 | T→I R@1 | T→I R@5 | T→I R@10 |
> |:--:|:--|:--:|:--:|:--:|:--:|:--:|:--:|
> | 0% | No pruning | 42.20 | 66.21 | 74.94 | 26.66 | 48.94 | 58.91 |
> | 12.5% | TE  |**40.01** |**64.26** | **73.88** | **25.69** | **47.72** | **57.86** |
> | 12.5% | SmartTrim | 38.54 | 62.54 | 71.99 | 24.63 | 46.26 | 56.30 |
> | 12.5% | Top-ViM | 27.52 | 51.43 | 62.56 | 18.16 | 37.78 | 48.12 |
> | 25% | TE  | **39.14** | **63.16** | **72.62** | **25.02** | **46.95** | **57.09** |
> | 25% | SmartTrim | 32.45 | 55.42 | 65.19 | 20.08 | 40.02 | 49.84 |
> | 25% | Top-ViM | 25.11 | 48.53 | 59.73 | 17.37 | 36.90 | 47.23 |
> | 37.5% | TE  | **32.45** | **56.80** | **66.92** | **20.86** | **41.92** | **52.25** |
> | 37.5% | SmartTrim | 30.47 | 53.32 | 63.21 | 19.22 | 38.40 | 48.11 |
> | 37.5% | Top-ViM | 24.10 | 47.20 | 58.48 | 15.99 | 34.95 | 45.12 |
> | 50% | TE  | **29.73** | **53.24** | **63.77** | **18.78** | **38.88** | **48.99** |
> | 50% | SmartTrim | 22.59 | 42.04 | 51.41 | 13.87 | 29.58 | 38.40 |
> | 50% | Top-ViM | 22.27 | 44.57 | 55.41 | 14.14 | 31.89 | 41.74 |
>
>
> **Table:** LLaVA-1.5-7B on VQAv2 under token pruning. We compare soft accuracy (%) of TE-based pruning with Top-ViM [2] and SCOPE [3] across pruning ratios from 0% to 60% for each layer in vision tower.
>
> | Pruning (%) | TE-based pruning | Top-ViM | SCOPE |
> |:-----------:|:----------------:|:-------:|:-----:|
> | 0  | 64.80 | 64.80 | 64.80 |
> | 20 | **64.67** | 57.73 | 62.03 |
> | 30 | **60.50** | 55.90 | 57.53 |
> | 40 | **59.80** | 42.07 | 47.07 |
> | 50 | **56.83** | 36.57 | 41.12 |
> | 60 | **56.07** | 34.27 | 36.27 |
>
> [1] Wang, Z., et al. SmartTrim: Adaptive tokens and attention pruning for efficient vision-language models. LREC-cOLING 2024.
>
> [2] Zhan, Z, et al. Exploring token pruning in vision state space models. NeurIPS 2024.
>
> [3] Deng, J., et al. SCOPE: Saliency-coverage oriented token pruning for efficient multimodel LLMs. NeurIPS 2025.

---

> > ### Author Rebuttal · Reviewer_AXcK · 2026-04-04
> >
> > This paper generally provides two central contributions
> >
> > First, it introduces, defines, and justifies the proposed Transfer Entropy (TE) metric. The metric is well explained and easy to understand. Second, it evaluates TE for the task of layer pruning.
> >
> > My concern with this approach is that the analysis is shallow relative to the implementation of the metric. In the original manuscript, the only results that tie the sophisticated measure of TE to real world performance are on layer pruning on 12 layer networks that are well understood (minus LLAVA-1.5-7B which is solely evaluated in the appendix and therefore by ICML reviewing standards outside of the scope of my original review, although I will address it in this followup). As the authors state in their rebuttal:
> >
> > "We therefore view the main novelty as a new directed-information lens for LLMs and VLMs, rather than only the pruning application."
> >
> > Therefore, this is not a pruning paper, but instead an information flow analysis paper. However, the pruning application is the only result that "ties down" the proposed metric toward being meaningful in reality. A significant amount of the work (and the abstract) is devoted to analyzing where the TE metric is higher or lower (Fig 1, Fig 2). *This analysis is meaningless unless TE is well connected to actual results, for which layer pruning on small networks is the only evaluation in the main text*. Furthermore, layer pruning is not a real task on its own, rather it is one of many techniques to make more compute efficient methods at inference.
> >
> > I am entirely willing to believe that TE is a great metric for analyzing generic machine learning models. However, the evidence in the text is not diverse enough to present this claim.
> >
> > *W1* Thank you for the results on a larger network, this strengthens the claims of the paper beyond small scale experiments where exhaustive layer search is possible for pruning.
> >
> > *W2* Thank you for pointing this out, I admit to misreading Equation 1.
> >
> > *W4* Your response does not address my concern about originality. I was never concerned that the originality of the proposed approach is only for pruning. Instead, the sufficiently novel TE method is not backed up by enough different results other than pruning to support the additional novelty.This paper provides a new measure for information flow. What information is flowing? All I see in the main text is scores for how much information is flowing, purely evaluated for accuracy on the task of layer pruning.
> >
> > *New tokenwise TE results*. Thank you for providing a new source of analysis for the pruning metric other than layer pruning. These new results are the reason I will increase my score to weak reject, as the provided score for information flow has now been evaluated on multiple axes. However, I stand by my earlier point that the results for this information flow metric are insufficient for analysis of where this metric activates to be useful. This paper is under-evaluated for its central claims of what transfer entropy is useful for, and this is the reason I still do not support acceptance.
> >
> > Please update the figures to be readable. Figure 1 is unacceptably small.
> >
> > Thank you for your response.

---

> > > ### Author Response · Authors · 2026-04-05
> > >
> > > Thank you for your detailed follow-up. To further address **W4**, we have added an application on attention-head pruning.
> > >
> > > In Appendix I.2, we analyze information flow among attention heads in CLIP.  We further use $TE_{\ell,h}$ as a criterion for attention-head pruning. Specifically, among the heads classified as benign, we prune those with lower $TE_{\ell,h}$ values. We compare our approach with SmartTrim, and the following Table shows that TE-based pruning consistently achieves better retrieval performance under matched pruning budgets. We also include an ablation in which we instead prune high-TE heads; the resulting performance drop, shown in the “Ablation: High-TE” row, further supports the importance of preserving high-TE heads. A high $TE_{\ell,h}$ indicates that head $h$ in layer $\ell$ has a strong directed functional impact on representation updates, whereas benign heads with low $TE_{\ell,h}$ are less important and are therefore more suitable pruning candidates.
> > >
> > >
> > > **Table:** MSCOCO retrieval performance of CLIP ViT-B/16 under attention head pruning, where $x$V+$y$T denotes pruning $x$ of the 12 attention heads per layer in the vision tower and $y$ of the 8 attention heads per layer in the text tower. The first and last layers are kept unpruned.
> > >
> > > | Pruning per Layer | Method | I2T R@1 | I2T R@5 | I2T R@10 | T2I R@1 | T2I R@5 | T2I R@10 |
> > > |:--|:--|--:|--:|--:|--:|--:|--:|
> > > | None | No Pruning | 42.20 | 66.21 | 74.94 | 26.66 | 48.94 | 58.91 |
> > > | 2V + 0T | TE | **39.00** | **63.53** | **73.18** | **24.84** | **47.13** | **57.34** |
> > > | 2V + 0T | SmartTrim | 37.25 | 61.65 | 71.42 | 23.69 | 45.61 | 55.86 |
> > > | 3V + 0T | TE | **37.56** | **61.98** | **71.89** | **24.16** | **46.04** | **56.27** |
> > > | 3V + 0T | SmartTrim | 36.25 | 60.48 | 70.38 | 22.95 | 44.62 | 54.84 |
> > > | 3V + 0T | Ablation: High-TE | 27.60 | 51.31 | 61.96 | 17.64 | 37.40 | 47.72 |
> > > | 3V + 2T | TE | **36.97** | **61.94** | **71.82** | **23.49** | **45.40** | **55.67** |
> > > | 3V + 2T | SmartTrim | 34.92 | 59.79 | 69.78 | 21.78 | 43.12 | 53.34 |
> > > | 5V + 3T | TE | **32.41** | **56.64** | **66.89** | **20.40** | **41.41** | **51.77** |
> > > | 5V + 3T | SmartTrim | 30.53 | 54.42 | 64.87 | 19.16 | 39.48 | 49.66 |
> > >
> > > Taken together, we hope the two rebuttals now tie TE to model behavior on multiple axes: layer pruning, token pruning, and attention-head pruning. More broadly, our claim is that TE measures directed information flow in representation updates across layers, heads, and tokens, rather than only producing abstract scores.
> > >
> > > TE is aimed at measuring update-driving influence in space (layers/heads/tokens) and time $t\rightarrow t+1$. This makes TE useful not only for identifying redundancy, but also for interpreting which parts of an LLM or VLM actively contribute to representation updates, for diagnosing bottlenecks, and potentially for guiding architecture or system design. For example:
> > >
> > > **1. Optimization diagnosis (Appendix H.3):** An inappropriate learning rate causes large epoch-to-epoch fluctuations in information flow, serving as a symptom of an unstable tuning process.
> > >
> > > **2. Fine-tuning stability (Appendix H.2):** By contrast, LoRA-based fine-tuning exhibits much smaller variation in information flow across epochs, consistent with the empirical stability of LoRA.
> > >
> > > **3. Architecture vs. dataset effects (Appendix H.5):** The architecture-versus-dataset study shows that model architecture primarily determines the overall shape of the information-flow profile, while datasets introduce only relatively small variations.
> > >
> > > **4. Architecture insight on larger LLMs (Appendix H.6):** The information-flow results on Llama-3.2-3B and Qwen-2.5-7B suggest that larger decoder-only LLMs share a common architecture-level pattern, with most task-relevant information flow concentrated in the middle layers.
> > >
> > > **5. Multimodal depth insight (Appendix I.1):** The vision$\to$text information-flow profile of LLaVA-1.5-7B provides insight into why this VLM requires many layers.
> > >
> > > **6. Head-level importance (Appendix I.2):** The attention-head information-flow analysis reveals the relative importance of different heads within each layer and shows that benign heads are generally less important.
> > >
> > > We appreciate the comment about readability on figures. We are sorry that Figure 1 is too small, and we will enlarge the subfigures if we have the opportunity to update the paper.
> > >
> > > Thank you again for your constructive comments, which have helped us better clarify the broader usefulness of TE-based information flow analysis.

---

### Official Review · Reviewer_Zdig · 2026-03-13

**Soundness:** 3
**Presentation:** 3
**Significance:** 3
**Originality:** 3
**Overall Recommendation:** 4
**Confidence:** 3

**Summary:**

This work introduces transfer entropy (TE) between layers to characterize information flow in LLMs/VLMs. With the perturbation-based approximation, TE can act as a principled and tractable measure of information propagation and architectural redundancy, and hence motivate guided model compression (in the layer level).

**Compliance With Llm Reviewing Policy:**

Affirmed.

**Final Justification:**

The rebuttal addressed my main concerns, and I will keep the positive recommendation.

**Key Questions For Authors:**

1. In Line 133 (right), what does curse of dimensionality represent? What quantities are exponential in dimensions?
2. For better readability, it seems better to provide a proof sketch to show why probabilities in TE definition are related to the log determinant form.
3. In Sec. 4.3, Why does large TE imply limited information gain and hence layer redundancy? It would be beneficial to include self-contained quantitative demonstrations.
4. In Eq. (12), by definition $z=(v, y)$ in LHS relates to $(J_l, 0)$ (Eq. (7) and Eq. (8)). However, $J_{l+1}$ appears in RHS. Where does this inconsistency come from?
5. In Eq. (21),  what is the definition of $S^{\text{null}}$? Do we need further ablations on $\lambda$ to check its effect on layer pruning?
6. For the mini-batch setting, do we require that TE is permutation-invariant? Eq. (18) seems not.

**Limitations:**

Yes.

**Strengths And Weaknesses:**

Strengths:
1. It is an interesting and potentially novel idea to measure architectural redundancy from an information flow viewpoint rather than traditional representation similarity.
2. The paper is well-organized and easy to follow, with reasonable logical development.
3. The derived entropy estimator is quantitative, tractable and lightweight.
4. The effectiveness of entropy-based model pruning is numerically verified through experiments.

Weaknesses:
1. The entropy estimator is derived by (stochastic) perturbation analysis, which is both local and linear. It is not clear to what extent this approximation holds, particularly for highly nonlinear networks. In addition, the derivation is only for Gaussian noises and hence noise-dependent, not covering varied perturbation directions.
2. In the current form, the estimation theory and model reduction experiments are mainly in a static manner. It is not clear how the efficiency gains are reflected given varied models and datasets in practice. Specifically, what is the regime or pipeline to apply this entropy -based model pruning method in practical applications to obtain real training accelerations? Do we need a dynamic adaption or model reuse across different tasks?
3. Applying this entropy estimator in mini-batches introduces additional computation overheads, which can be significant in the large-scale parallel setting in practice.

---

> ### Author Rebuttal · Authors · 2026-03-29
>
> We are grateful for the reviewer’s valuable comments and suggestions. We address the concerns below.
>
> **W1**  We agree that our estimator is a local linear–Gaussian approximation, not an exact TE computation for the full nonlinear model. This is already stated in the paper: exact TE is intractable in high-dimensional LLM/VLM activations, and the cosine estimator is used primarily as a relative ranking/pruning signal rather than an exact TE value. Appendix D further provides a conservative error bound for the trace surrogate and discusses stability via EMA smoothing and null-baseline subtraction. Appendix G gives an illustrative example showing that the approximation tracks the exact TE trend in practice. Regarding Gaussian noise, it is used as a tractable isotropic probe; in practice, Theorem 4.3 replaces $J\delta x$ with finite differences across batch samples/tokens, so the proxy is not restricted to a single perturbation direction.
>
> **W2** Our current method is evaluated in a static prune-once pipeline: estimate TE during a short warm-up / fine-tuning stage, smooth and calibrate the scores, prune high-TE layers with guardrails, and then do a brief recovery fine-tune. The practical gain is therefore lower downstream training/inference cost after pruning, rather than a claim of fully dynamic acceleration throughout the entire original training run. We do not require dynamic adaptation or cross-task model reuse for the current method; those are natural future extensions, and the paper already mentions online usage such as adaptive depth and early exiting as future work.
>
> **W3** We agree that mini-batch TE estimation introduces additional overhead. In the paper, we report TE at every epoch mainly to allow readers to observe its dynamics throughout training. In practical use, however, the method does not require TE to be estimated at every epoch. Instead, as described in Section 4.3, TE can be computed only occasionally, for example, over one epoch or a sliding window during a short warm-up stage before pruning.
>
> **Q1** Here, the 'curse of dimensionality' refers to the difficulty of estimating the high-dimensional joint and conditional densities required by exact TE. The TE value itself is not exponential in dimension. Rather, the quantities that grow exponentially are those associated with nonparametric density estimation, such as the number of discretization cells and the corresponding sample complexity needed to cover the joint space. For example, in CLIP ViT-B/16, the vision embedding size is d=768. Since the exact TE term involves three d-dimensional variables, the joint space has dimension 3d=2304. If each dimension is discretized into m=10 bins, an exact nonparametric TE estimator would require $10^{2304}$ cells, which is clearly intractable.
>
> **Q2** The detailed proof is provided in Appendix B, specifically in lines 656–698. A proof sketch could be:  TE definition can be rewritten as conditional mutual information, then specialized under the local Jacobian Gaussian approximation so that the conditional entropies reduce to log-determinants of covariance matrices; applying the Schur complement then gives the log-determinant form in Theorem 4.1.
>
> **Q3** The intuition is that $TE_l(t)=I(X_{l+1}^{t+1};X_l^{t+1}|X_{l+1}^t)$ measures how much of the updated representation at layer $l+1$ is already explained by layer $l$, after controlling for the previous state of layer $l+1$. Therefore, a large $TE_l(t)$ indicates limited additional information gain at layer $l+1$ and hence greater redundancy. This is supported quantitatively by our reverse-TE pruning results in Table 1: pruning high-TE layers stays close to the baseline, while pruning low-TE layers leads to substantial degradation.
>
> **Q4** Thanks for pointing this out. Since $y \approx y_0$ in Eq. (8), for $z=(v,y)$, the correct form of eqn (12) is $\Sigma_{zz}=\begin{pmatrix}\Sigma_{vv} & 0 \\\\ 0 & 0\end{pmatrix}=\sigma^2\begin{pmatrix}J_\ell J_\ell^\top & 0 \\\\ 0 & 0\end{pmatrix}$, which is singular. Therefore, $\Sigma_{zz}^{-1}$ in our derivation should be understood as a Moore-Penrose pseudoinverse.
>
> **Q5** $S^{\text{null}}_{\ell+1}$ is defined in Lines 226-230 as the null TE baseline. We also performed ablations on $\lambda$. In RoBERTa on MNLI, the pruning order at $\lambda=0$ is $[9,10,8,11,5,6,7,4,3,2,12]$; at $\lambda=0.5$, only 8 and 11 switch positions. In RoBERTa on MRPC, the pruning order at $\lambda=0$ is $[8,6,5,10,9,7,11,4,3,2,12]$; at $\lambda=0.5$, only 7 and 11 switch positions. Overall, this suggests that the layer ranking is reasonably stable under moderate changes in $\lambda$.
>
> **Q6** Eq. (18) is not strictly permutation-invariant at the single mini-batch level. In our implementation, however, the train dataloader uses random reshuffling, which changes these local pairings across batches and helps reduce dependence on any particular fixed ordering. Averaging over batches, this yields a more stable epoch-level TE estimate.

---

> > ### Author Rebuttal · Reviewer_Zdig · 2026-04-02
> >
> > Authors have adequately addressed raised concerns. I will keep my recommendation.

---

> > > ### Author Response · Authors · 2026-04-05
> > >
> > > Thank you for confirming that your concerns have been addressed. We appreciate your continued support.

---

### Official Review · Reviewer_PE3r · 2026-03-16

**Soundness:** 3
**Presentation:** 3
**Significance:** 3
**Originality:** 2
**Overall Recommendation:** 4
**Confidence:** 3

**Summary:**

This paper is concerned with measuring directed information flow through LLMs and VLMs, using the transfer entropy as the measure to track between layers.  This enables things like localizing where multimodal fusion is happening and thereby suggesting alternative (presumably more efficient) architectures.  In general, this is meant as a pretty broad diagnostic and interpretability approach.

**Compliance With Llm Reviewing Policy:**

Affirmed.

**Key Questions For Authors:**

1. Please do explain distinction from "DeepInsert"/"Skip It".if you can.  What does directed information flow buy you beyond classical information measures.

2. Please explain the relationship between the Massey-Mitter-Tatikonda directed information and the transfer entropy: it would be good to know if there are coding theorems that provide the transfer entropy with operational significance for some task.

3. Can you say more about any architectural insights the directed information flow analysis gives?

**Limitations:**

yes

**Strengths And Weaknesses:**

The logic of using causal information measures is sound and the results are interesting, especially about finding where the multimodal fusion happens in an information-theoretic sense. On the experiment side, all appears to be sound, and in particular the reported experiments are for a variety of architectures and settings which suggests generality.

The paper;s organization is good so it makes sense.  No particular comments on the presentation.

The understanding gained from causal information-theoretic measures seems to be good, and it seems this general approach could be widely applicable for other interpretability settings.

I am concerned about originality.  DeepInsert (https://arxiv.org/abs/2504.19327) and its followon work "Skip It" (https://arxiv.org/abs/2509.25584) seem to be fairly similar in terms of findings, so would be helpful to understand the relationship between what is found with traditional information measures vs. causal ones.

---

> ### Author Rebuttal · Authors · 2026-03-30
>
> We thank the reviewer for the helpful and detailed feedback, which has helped us clarify several aspects of the paper.
>
> **Q1/W1** Our work differs from DeepInsert and Skip-It in that those papers primarily propose efficiency interventions, whereas ours is a more general measurement framework. DeepInsert introduces a specific architectural modification for late multimodal token insertion, and Skip-It develops criteria for when layer skipping is safe in VLMs. In particular, Skip-It uses the conditional entropy $H(X_\ell |X_{\ell-1})$ as a notion of informational redundancy. Since $H(X_\ell |X_{\ell-1})=H(X_\ell)-I(X_\ell; X_{\ell-1})$, a low conditional entropy implies high mutual information (redundancy) between adjacent layers. However, this remains an essentially mutual-information-based view: it does not exclude the previous state of layer $\ell$, and therefore cannot isolate the new information flow from layer $\ell-1$ to layer $\ell$. By contrast, our paper introduces transfer entropy as a directed, conditional measure of information flow across layers, heads, and modalities in both LLMs and VLMs. Although our practical proxy in Eq. (18) uses a function of cosine similarity, that cosine term is not used as a standalone undirected similarity measure; it is derived as a surrogate for the asymmetric conditional quantity $TE_\ell(t)=I\left(X_{\ell+1}^{t+1};X_\ell^{t+1}\mid X_{\ell+1}^{t}\right)$. Thus, the key advantage over classical measures such as cosine similarity, Centered Kernel Alignment (CKA), or mutual information is the target being measured: TE asks whether layer $\ell$ provides new predictive information about the update of layer $\ell+1$, beyond what is already explained by layer $\ell+1$'s own previous state, rather than merely whether two layers are similar or statistically associated. This is why our method can distinguish directed update influence from static similarity, which is exactly the signal we use to study redundancy, cross-modal fusion, and pruning.
>
> **Q2** Schreiber transfer entropy (TE) and Massey-Mitter-Tatikonda directed information (DI) are closely related but not identical. A one-step TE is
> $T_{X\to Y}=I(Y_{t+1};X_t\mid Y_t)$,
> whereas DI over a length-$n$ trajectory is
> $I(X^n\to Y^n)=\sum_{i=1}^n I(X^i;Y_i\mid Y^{i-1})$.
> Thus, DI uses the entire causal source history, while TE is a local one-step conditional mutual information. For stationary processes, the DI rate decomposes into a transfer-entropy-rate term plus an instantaneous-information-exchange term, so TE can be viewed as a local component of the broader DI framework. The main coding theorems are for DI: in particular, DI characterizes feedback capacity for broad classes of channels with memory/feedback. By contrast, in our setting TE is used because it directly captures whether one layer provides new predictive information about the next state of another layer beyond the receiver's own previous state, which is the local learning-dynamics quantity of interest here.
>
> **Q3** Our directed information-flow analysis gives several concrete architectural insights. First, it reveals that different Transformer families use depth in systematically different ways: in encoder-only RoBERTa, TE peaks in the middle layers, suggesting that task-dependent computation is concentrated there, while early and late layers behave more like input/output adapters; in decoder-only LLMs such as Llama and Qwen, TE is concentrated in a broader mid-depth band; and in encoder–decoder T5, the encoder and decoder exhibit clearly different TE patterns, indicating asymmetric functional roles across the two stacks. Second, in multimodal architectures, TE identifies where fusion and redundancy occur: in CLIP, redundancy concentrates in late vision blocks while the text tower maintains broadly elevated TE, whereas in LLaVA the vision$\to$text TE increases toward upper decoder layers, indicating that multimodal fusion is localized near the top of the language stack rather than spread uniformly across depth. Third, at the attention head level, combining TE with CLS/EOT attention mass yields role-aware head maps, which help distinguish sink/broadcaster-like responsibilities across depth. Overall, these findings go beyond pruning alone: they suggest how different architectures allocate computation across depth and modality, and can inform design choices such as adaptive depth, late fusion, and targeted placement of adapters or LoRA modules.

---

> > ### Author Rebuttal · Reviewer_PE3r · 2026-04-03
> >
> > Thanks to the authors for the responses.  For the first Q, understand the differences between the two, but still not quite sure what more the directed viewpoint buys you beyond the undirected mutual information in terms of architectural insights or other things one can do from this analysis.  It would also be helpful to include discussion of differences as well as what they buy you in the updated paper.

---

> > > ### Author Response · Authors · 2026-04-05
> > >
> > > Thank you for this helpful follow-up. The directed view point buys additional architectural insight because it reveals how information is propagated from one layer to the next, into and out of attention heads, and into and out of tokens, rather than only showing that these representations are correlated. Undirected MI is well suited to measuring association, while TE is aimed at measuring update-driving influence in space (layers/heads/tokens) and time ($t \rightarrow t+1$).
> > >
> > > This distinction is useful beyond analysis alone. Because TE identifies where new information is actually being transmitted, as opposed to where computation is largely redundant, it provides a principled signal for pruning layers, attention heads, and tokens. More broadly, it offers a dynamic view of which parts of an LLM/VLM are actively contributing to representation updates, which can improve interpretability and may also help with debugging, diagnosing bottlenecks, and guiding architecture or system design. For example:
> > >
> > > **1. Optimization diagnosis (Appendix H.3):** An inappropriate learning rate causes large epoch-to-epoch fluctuations in information flow, serving as a symptom of an unstable tuning process.
> > >
> > > **2. Fine-tuning stability (Appendix H.2):** By contrast, LoRA-based fine-tuning exhibits much smaller variation in information flow across epochs, consistent with the empirical stability of LoRA.
> > >
> > > **3. Architecture vs. dataset effects (Appendix H.5):** The architecture-versus-dataset study shows that model architecture primarily determines the overall shape of the information-flow profile, while datasets introduce only relatively small variations.
> > >
> > > **4. Architecture insight on larger LLMs (Appendix H.6):** The information-flow results on Llama-3.2-3B and Qwen-2.5-7B suggest that larger decoder-only LLMs share a common architecture-level pattern, with most task-relevant information flow concentrated in the middle layers.
> > >
> > > **5. Multimodal depth insight (Appendix I.1):** The vision$\to$text information-flow profile of LLaVA-1.5-7B provides insight into why this VLM requires many layers.
> > >
> > > **6. Attention head importance (Appendix I.2):** The attention-head information-flow analysis reveals the relative importance of different heads within each layer and shows that benign heads are generally less important.
> > >
> > > In this sense, the value of the directed viewpoint is not simply that it is asymmetric, but that it turns "these components are related" into the more actionable statement, "this component is driving the next update."
> > >
> > > We will include the differences and what they buy us if we have the opportunity to update the paper.

---

### Decision · Program_Chairs · 2026-04-30

**Decision:**

Reject

**Comment:**

The paper proposes using Transfer Entropy (TE) of directional information flow to discover dependencies between layers in LLMs and VLMs. Specifically, small perturbations are introduced in representations across layers to detect minor changes in activations that can calculate TE empirically with theoretical justifications. As such, TE can act as a principled and tractable measure of information propagation and architectural redundancy, which the authors use to guide layer-based model compression, pruning layers according to the metric.

Reviewers have appreciated the theoretical justifications of TE, and found the idea interesting, the experiments convincing, and the overall work solid. There has been a disagreement among the reviewers regarding the framing versus the main experiments of this work. Specifically, one reviewer argued that while the authors position the focus of their work on information flow, there is not enough substance on information flow and the pruning experiments are only a downstream application of it. The authors have attempted to address this concern through additional experiments during the rebuttal, but these only partially addressed the reviewer’s concern. Other reviewers view the title-content inconsistency as reasonable and believe the findings are already interesting.

Other than that, there were concerns regarding the generalization of the findings as the experiments focus on a rather small scale of LMs, but the authors have addressed that by extending their evaluations. Also, reviewers indicated the work could benefit from better contextualization with existing literature.

Overall, while this work makes interesting contributions and most reviewers were positive about its findings, due to its weaknesses (mostly around its framing), it cannot be accepted at the conference this time.